# Targeting DNA2 overcomes metabolic reprogramming in multiple myeloma

Natthakan Thongon[1], Feiyang Ma [2], Natalia Baran [1], Pamela Lockyer[1], Jintan Liu [3], Christopher Jackson[1], Ashley Rose[1], Ken Furudate [1], Bethany Wildeman[1], Matteo Marchesini[4], Valentina Marchica[5], Paola Storti [5], Giannalisa Todaro[5], Irene Ganan-Gomez [1], Vera Adema [1], Juan Jose Rodriguez-Sevilla[1], Yun Qing[6], Min Jin Ha[6], Rodrigo Fonseca [7], Caleb Stein[7], Caleb Class [8], Lin Tan[9], Sergio Attanasio[3], Guillermo Garcia-Manero [1], Nicola Giuliani [5], David Berrios Nolasco[10], Andrea Santoni[1], Claudio Cerchione[4], Carlos Bueso-Ramos [11], Marina Konopleva[1], Philip Lorenzi [9], Koichi Takahashi [1], Elisabet Manasanch[10], Gabriella Sammarelli[5], Rashmi Kanagal-Shamanna [11], Andrea Viale[3], Marta Chesi [7] & Simona Colla [1] ✉

DNA damage resistance is a major barrier to effective DNA-damaging therapy in multiple myeloma (MM). To discover mechanisms through which MM cells overcome DNA damage, we investigate how MM cells become resistant to antisense oligonucleotide (ASO) therapy targeting Interleukin enhancer binding factor 2 (ILF2), a DNA damage regulator that is overexpressed in 70% of MM patients whose disease has progressed after standard therapies have failed. Here, we show that MM cells undergo adaptive metabolic rewiring to restore energy balance and promote survival in response to DNA damage activation. Using a CRISPR/Cas9 screening strategy, we identify the mitochondrial DNA repair protein DNA2, whose loss of function suppresses MM cells' ability to overcome ILF2 ASO−induced DNA damage, as being essential to counteracting oxidative DNA damage. Our study reveals a mechanism of vulnerability of MM cells that have an increased demand for mitochondrial metabolism upon DNA damage activation.

The prevalence of multiple myeloma (MM), already the second most common hematological malignancy worldwide, will grow by almost 60% by 2030, making the disease an increasingly important public health challenge[1]. In the last decade, MM patients' clinical outcomes have significantly improved owing to the introduction of novel agents, which have doubled these patients' overall median survival duration. However, the expected survival duration for patients with higher-risk disease is still only about 2–3 years[2], likely because available agents

[1]Department of Leukemia, The University of Texas MD Anderson Cancer Center, Houston, TX, USA. [2]Division of Rheumatology, Department of Internal Medicine, Michigan Medicine, University of Michigan, Ann Arbor, MI, USA. [3]Department of Genomic Medicine, The University of Texas MD Anderson Cancer Center, Houston, TX, USA. [4]IRCCS Instituto Romagnolo per lo Studio dei Tumori (IRST) Dino Amadori, Meldola, Italy. [5]Department of Medicine and Surgery, University of Parma, Parma, Italy. [6]Department of Biostatistics, The University of Texas MD Anderson Cancer Center, Houston, TX, USA. [7]Department of Medicine, Mayo Clinic, Scottsdale, AZ, USA. [8]Department of Pharmaceutical Sciences, College of Pharmacy and Health Sciences, Butler University, Indianapolis, IN, USA. [9]Metabolomics Core Facility, Department of Bioinformatics and Computational Biology, The University of Texas MD Anderson Cancer Center, Houston, TX, USA. [10]Department of Lymphoma and Myeloma, The University of Texas MD Anderson Cancer Center, Houston, TX, USA. [11]Department of Hemopathology, The University of Texas MD Anderson Cancer Center, Houston, TX, USA. ✉e-mail: scolla@mdanderson.org

were developed without a clear understanding of the pathobiology underlying this aggressive phenotype.

The 1q21 gain/amplification which occurs in approximately 30% of de novo MMs, is among the most frequent chromosomal aberrations in MM patients and is considered a very high-risk genetic feature related to disease progression and drug resistance[3]. The 1q21 gain/amplification can be detected in up to 70% of patients as they develop relapsed and then refractory disease, likely because of the positive selection of a plasma cell clone that previously made up a minor fraction of the tumor bulk and/or the acquisition of new genetic alterations due to genomic instability. Among patients with the 1q21 gain/amplification who relapsed, the median overall survival duration is a dismal 9 months[4-6]. More recently, MM patients with at least 4 copies of 1q21, also defined as patients with "double hit", have a dire prognosis despite modern therapies and should be considered for novel therapeutic approaches[7].

In our previous studies, we identified the interleukin enhancer binding factor 2 gene, *ILF2*, as a key modulator of the DNA repair pathway in MM. ILF2 overexpression, driven by 1q21 copy number alterations, promotes adaptive responses to DNA damage in a dose-dependent manner, which explains why MM patients with the 1q21 gain/amplification benefit less from high-dose chemotherapy than patients without the gain/amplification. Mechanistically, copy-number-driven ILF2 levels promoted resistance to genotoxic agents by modulating mRNA processing and stabilization of transcripts involved in DNA repair pathways in response to DNA damage[8,9]. These results supported the development of strategies for blocking ILF2 signaling to enhance the effectiveness of current therapeutic approaches based on DNA-damaging agents in 1q21-amplified MM.

Here, we used antisense oligonucleotides (ASOs) to discover novel mechanisms through which MM cells overcome DNA damage activation and become resistant to therapeutic interventions affecting DNA repair pathways.

## Results

### ILF2 ASOs induce DNA damage activation and enhance MM cells' sensitivity to DNA-damaging agents

To deplete *ILF2* in 1q21 MM cells, we developed ILF2 ASOs with constrained ethyl chemistry, which induces improved stability, RNA affinity, and resistance against nuclease-mediated metabolism, resulting in a significantly improved tissue half-life in vivo and a longer duration of action[10,11].

To identify potential toxicities that could arise from ILF2 inhibition in healthy tissues, we injected ASOs targeting mouse *Ilf2* into male Balb/c mice (Supplementary Table 1). We did not observe either consistent histopathological or biochemical ASO-induced alterations, which suggests that Ilf2 depletion does not induce on-target toxicity (Fig. 1A).

We then screened about 300 ASOs targeting human *ILF2* and performed a dose-response confirmation for the 5 most effective ILF2 ASOs in the MM cell line JJN3. The ILF2 ASO 1146809 (09), which elicited the best dose response and had an acceptable tolerability profile in mice was selected for functional validation studies in MM cells (Supplementary Fig. 1A, B, and Supplementary Table 1). To determine the biological effect of ILF2 ASOs on MM cells with the 1q21amplification, we treated the 1q21 amplified MM cell lines JJN3 and KMS11[12] with increasing concentrations of non-targeting (NT) ASOs and ILF2 ASOs. We observed that ILF2 depletion was associated with significant γH2AX foci accumulation (Fig. 1B), apoptosis (Supplementary Fig. 1C), and inhibition of cell proliferation (Supplementary Fig. 1D), which is consistent with our previous findings using shRNAs targeting *ILF2*[8].

To determine the role of ILF2 in the regulation of the DNA damage response in MM cells, we evaluated whether ASO-mediated ILF2 depletion increased MM cells' sensitivity to DNA-damaging agents routinely used in the treatment of MM. Employing melphalan to induce

DNA double-strand breaks, we found that ILF2 ASO−treated MM cells exposed to melphalan for 6 h had increased γH2AX induction and caspase 3 activation as compared with NT ASO−treated MM cells exposed to melphalan (Fig. 1C). These results aligned with the significant increase in the number of annexin+ ILF2 ASO−treated MM cells that we observed when the treatment with melphalan was extended to 48 h (Supplementary Fig. 1E). We also observed that ILF2 depletion sensitized MM cells to bortezomib (Fig. 1D; Supplementary Fig. 1F), which is consistent with previous findings showing that bortezomib impairs homologous recombination[13], thus enhancing the effect of ILF2 depletion on the ability of MM cells to repair DNA damage[8]. Similar data were obtained using the MM cell lines MM1R (Supplementary Fig. 1G, H, I), H929 (Supplementary Fig. 1J, K), and RPMI-8226 (Supplementary Fig. 1L, M), which harbor 1q21 gain or amplification.

To validate the effectiveness of ILF2 ASOs in enhancing the effect of DNA-damaging agents in vivo, we established a MM xenograft model that recapitulates the disseminated nature of MM and the features of its bone and organ metastases in humans. To this end, we transduced KMS11 cells with a lentiviral vector delivering the green fluorescent protein (GFP) ZsGreen and the luciferase reporter transgene to create GFP+Luc+ KMS11 cells, which were injected via the tail vein into sublethally irradiated NSG mice. The mice were randomized based on the level of tumor burden detected by bioluminescence imaging and injected daily with NT or ILF2 ASOs for 7 days. To evaluate whether ILF2 ASOs sensitized MM cells to DNA-damaging agents, we further treated the xenografts with NT or ILF2 ASOs every other day in combination with melphalan and evaluated tumor burden at the end of the third cycle of the combination therapy (Supplementary Fig. 1N). Immunohistochemical analysis showed a 50% reduction in ILF2 levels in KMS11 cells from the bone marrow (BM) and the liver of xenografts treated with ILF2 ASOs in combination with melphalan. These data were confirmed by real-time PCR in GFP+KMS11 cells isolated from the xenografts (Supplementary Fig. 1O). Consistent with these results, ILF2 depletion was associated with increased levels of caspase 3 activation (Supplementary Fig. 1P) and reduced tumor burden (Fig. 1E). These data suggest that even a 50% reduction in MM cells' ILF2 levels enhances the anti-tumor effect of melphalan on MM cells in vivo.

### Metabolic reprogramming mediates MM cells' resistance to DNA damage activation

DNA damage resistance is a major barrier to effective DNA-damaging therapy in MM. To evaluate whether MM cells could eventually become resistant to the DNA damage induced by ILF2 depletion, we treated JJN3, KMS11, MM1R, H929, and RPMI-8226 cells with NT or ILF2 ASOs for more than 3 weeks. Whereas KMS11 (Fig. 2A), MM1R, H929, and RPMI-8226 (Supplementary Fig. 2A) maintained high levels of DNA damage activation and had significantly increased rates of apoptosis after 3 weeks of ILF2 ASO, JJN3 cells overcame ILF2 ASO−induced DNA damage activation and became resistant to ILF2 ASO treatment (Fig. 2B), which suggests that MM cells can eventually activate compensatory mechanisms to overcome the deleterious effects of DNA damage and maintain their survival.

To gain insights into the molecular mechanisms by which MM cells overcome ILF2 ASO−induced DNA damage activation, we performed bulk RNA sequencing (RNA-seq) analysis of ASO-treated KMS11 and JJN3 cells at early (1 week) and late (3 weeks) treatment time points (Supplementary Fig. 2B). We observed that most of the genes that were significantly downregulated in JJN3 cells (but not KMS11 cells) treated with ILF2 ASOs for more than 3 weeks, as compared with those treated for 1 week, were involved in the regulation of the DNA damage response (Supplementary Fig. 2C). To exclude the possibility that continuous ILF2 ASO exposure could lead to the selection of MM clones intrinsically resistant to ILF2 ASO−induced DNA damage, we performed single-cell RNA-seq (scRNA-seq) analysis of JJN3 cells treated with NT or ILF2 ASOs for 3 weeks (Supplementary Fig. 2D). Our

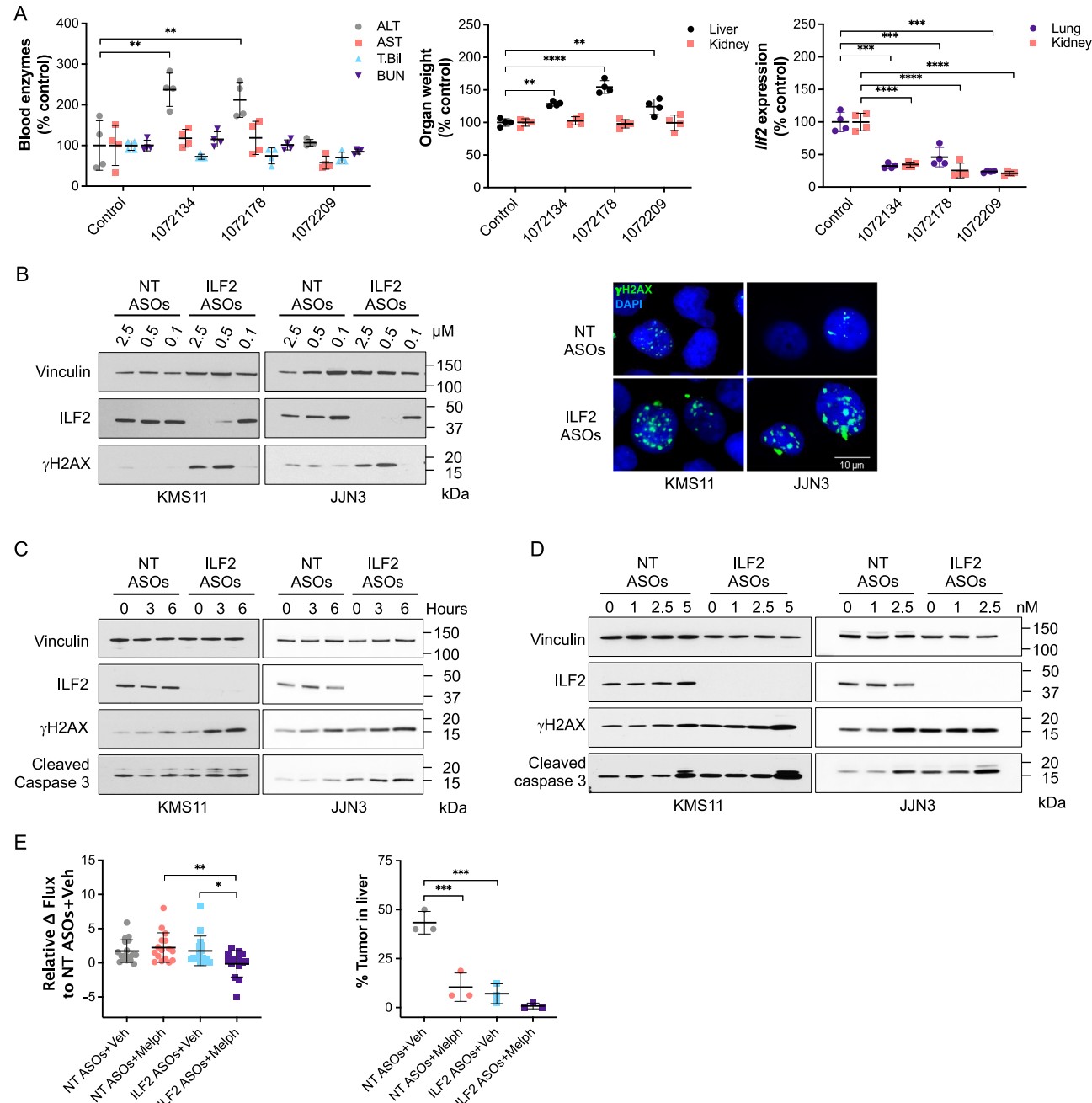

analysis divided JJN3 cells into 2 main clusters that were independent of treatment (Fig. 2C; Supplementary Fig. 2E). Further copy number alteration analysis demonstrated that persistent exposure to ILF2 ASOs did not induce clonal selection (Supplementary Fig. 2F). Differential gene expression analysis of NT ASO– or ILF2 ASO–treated cells in each of these clusters revealed that the significantly upregulated genes in ILF2 ASO–treated cells were mainly involved in oxidative phosphorylation (OXPHOS), mTORC1 pathway, DNA repair signaling, cell cycle regulation, and reactive oxidative species (ROS; Fig. 2D; Supplementary Fig. 2G).

To validate these findings, we evaluated the metabolomic changes induced by long-term exposure to ILF2 ASOs. Our targeted metabolomic analysis showed that among the 33 metabolites that were increased in ILF2 ASO–resistant cells, intermediates in the tricarboxylic acid cycle and pyrimidine pathways were significantly enriched ($P = 0.016$ and $P < 0.001$, respectively; Fig. 2E; Supplementary Fig. 2H). Consistent with this observation, ILF2 ASO–resistant JJN3 cells were

significantly more sensitive to the OXPHOS inhibitor IACS-010759[14] than the ILF2 ASO–sensitive cells were (Supplementary Fig. 2I). In contrast, the pyrimidine inhibitor brequinar[15] could not overcome MM cells' resistance to ILF2 ASO–induced apoptosis (Supplementary Fig. 2I). ILF2 ASO–resistant cells had significantly higher maximal oxidative consumption rates (OCRs) than NT ASO–treated cells did, which suggests that these cells' have a higher adaptation capability. Compared with NT ASO–treated cells exposed to IACS-010759, ILF2 ASO–treated cells exposed to IACS-010759 had significantly lower OCRs (Supplementary Fig. 2J) and higher mitochondrial ROS production (Supplementary Fig. 2K). To evaluate whether OXPHOS inhibition could efficiently target MM cells in vivo, we established a MM xenograft model by transplanting ILF2 ASO–resistant GFP+Luc+ JJN3 cells into NSG mice. Mice were treated with NT or ILF2 ASOs in the presence or absence of IACS-010759 (Supplementary Fig. 2L). Consistent with our hypothesis, ILF2 ASO–treated mice that received IACS-010759 had

**Fig. 1 | ILF2 ASOs induce DNA damage activation and enhance MM cells' sensitivity to DNA-damaging agents. A** Left, levels of alanine aminotransaminase (ALT), aspartate aminotransaminase (AST), total bilirubin (T. Bil), and blood urea nitrogen (BUN) in the peripheral blood of Balb/c mice treated with phosphate-buffered saline (control; $n = 4$) or one of 3 different ASOs targeting *Ilf2* ($n = 4$ per each ASO). **$P < 0.01$; control vs 1072134: $P = 0.0018$; control vs 1072178: $P = 0.0078$. Middle, relative weights of the liver and kidneys in each mouse. ****$P < 0.0001$; **$P < 0.01$; control vs 1072178: $P < 0.0001$; control vs 1072134: $P = 0.0011$; control vs 1072209: $P = 0.004$. Right, relative *Ilf2* expression in the kidneys and lungs of the mice. Statistically significant differences were detected using one-way ANOVA (****$P < 0.0001$; ***$P < 0.001$). The mean ± S.D. is shown. **B** Left, Western blot analysis of ILF2 and γH2AX in KMS11 (left) and JJN3 (right) cells treated with NT or ILF2 ASOs at the indicated concentrations for 1 week. Vinculin was used as the loading control. Right, anti-γH2AX immunofluorescence in KMS11 (left) and JJN3 (right) cells treated with NT or ILF2 ASOs (0.5 and 1 μM, respectively) for 1 week. Green indicates γH2AX; blue, DAPI. Scale bars represent 10 μm. Two biological replicates were performed. **C** Western blot analysis of ILF2, γH2AX, and cleaved caspase 3 in KMS11 (left) and JJN3 (right) cells treated with NT or ILF2 ASOs (0.5 and 1 μM, respectively) for 1 week prior to exposure to 10 μM melphalan for 0, 3, and 6 h. Vinculin was used as the loading control. Two biological replicates were

performed. **D** Western blot analysis of ILF2, γH2AX, and cleaved caspase 3 in KMS11 (left) and JJN3 (right) cells treated with NT or ILF2 ASOs (0.5 and 1 μM, respectively) for 1 week prior to receiving bortezomib for 48 h at the indicated concentrations. Vinculin was used as a loading control. Three biological replicates were performed. **E** Left, differences in the luciferase signal in NSG mice engrafted with GFP⁺Luc⁺ KMS11 cells after receiving NT or ILF2 ASOs for 1 week and NT or ILF2 ASOs with vehicle (Veh) or in combination with melphalan (Melph) every other day for 5 more days. Data are expressed as the mean bioluminescence activity relative to that of the NT ASOs+Veh group from each mouse [Δ flux of luciferase signal (photons/second, p/s)] ±S.D. (NT ASOs+Veh, $n = 17$; NT ASOs+Melph, $n = 16$; ILF2 ASOs+Veh, $n = 16$; ILF2 ASOs+Melph; $n = 14$ from 2 independent experiments). Statistically significant differences were detected by one-way ANOVA (**$P < 0.01$; *$P < 0.05$; NT ASOs+Melph vs ILF2 ASOs+Melph: $P = 0.0072$; ILF2 ASOs+Veh vs ILF2 ASOs+Melph: $P = 0.0465$). Right, tumor burden in the liver of the xenografts at day 12 of treatment. Data are expressed as percentages calculated by dividing the tumor area by the total area of the liver. The mean ± S.D. for 3 representative mice per group are shown. Statistically significant differences were detected by one-way ANOVA (***$P < 0.001$; NT ASOs+Veh vs NT ASOs+Melph: $P = 0.0003$; NT ASOs+Veh vs ILF2 ASOs+Veh: $P = 0.0001$). Source data are provided as a Source Data file.

a significantly longer survival duration than those that did not receive IACS-010759 ($P = 0.0006$; Supplementary Fig. 2M).

Together, these data suggest that MM cells can undergo an adaptive metabolic rewiring to restore energy balance and promote cell survival in response to DNA damage activation.

## DNA2 is essential for maintaining MM cells' survival after DNA damage−induced metabolic reprogramming

We hypothesized that ILF2 ASO−resistant cells' metabolic reprogramming relied on the repair of DNA damage induced by either ILF2 depletion or by the generation of ROS from activated mitochondrial metabolism and that targeting DNA repair proteins involved in these processes could overcome MM cells' resistance to DNA damage. To test this hypothesis, we used a CRISPR/Cas9 library screening strategy to identify DNA repair genes whose loss of function could suppress MM cells' capability to overcome resistance to ILF2 ASO−induced DNA damage. To this end, we designed a library of pooled single-guide RNAs (sgRNAs) targeting 196 genes involved in DNA repair pathways and DNA damage response regulation and cloned these sgRNAs into the pLentiGuide-Puro lentiviral vector (Supplementary Data 1). We infected Cas9-transduced JJN3 and KMS11 cells using a multiplicity of infection <0.3 to ensure that each MM cell was transduced by only 1 sgRNA. A representative portion of the total cells was collected 48 h after the transduction and used as a reference sample. Cells were selected with puromycin and treated with NT or ILF2 ASOs for 3 weeks before collection (Fig. 3A; Supplementary Fig. 3A). To identify ILF2 ASO sensitizer genes (genes whose sgRNAs were negatively selected in only ILF2 ASO−treated cells), we used deep sequencing of the sgRNA barcodes and the drugZ algorithm[16] to assess differences in the representation of all sgRNAs between NT ASO− and ILF2 ASO−treated cells across the 3 sets of experiments (Supplementary Fig. 3B). As expected, sgRNAs targeting essential genes were depleted in both NT ASO− and ILF2 ASO−treated JJN3 and KMS11 cells (Supplementary Fig. 3C). Compared with those in NT ASO−treated cells, sgRNAs targeting *MMS19*, *DNA2*, and *DDB1* were significantly depleted in ILF2 ASO−treated JJN3 cells but not in KMS11 cells after 3 weeks of treatment ($P < 0.01$; Fig. 3B; Supplementary Fig. 3D), suggesting that the MMS19, DNA2, and DDB1 repair proteins may have roles in promoting resistance to ILF2 depletion.

Among these 3 DNA repair proteins, the nuclease/helicase DNA2, which is localized in the mitochondria but not in the nuclei of MM cells (Fig. 3C and Supplementary Fig. 3E), was the only druggable target[17]. Higher levels of *DNA2* expression were correlated with 1q21 amplification (Supplementary Fig. 3F) and poorer overall survival in MM

patients treated with velcade, revlimid, and dexamethasone or high-dose chemotherapy followed by tandem autologous transplantation (Fig. 3D), proteasome inhibitors (PIs) alone or in combination with other therapies but not in those treated with immunomodulatory drugs (Supplementary Fig. 3G). Based on these correlative observations, we hypothesized that targeting DNA2 ultimately overcomes DNA damage-induced metabolic reprogramming.

To test this hypothesis, we used the specific DNA2 activity inhibitor NSC105808 (NSC)[18]. We confirmed that targeting DNA2 activity overcame resistance to ILF2 ASOs and induced MM cell death in vitro (Supplementary Fig. 3H) by inducing apoptosis (Fig. 3E). Importantly, NSC did not induce DNA damage in MM cells (Fig. 3F), which further confirms that DNA2 does not have a nuclear repair function in MM. Similar results were obtained using the DNA2 inhibitor C5[19] (Supplementary Fig. 3I). To evaluate whether DNA2 activity inhibition can efficiently target MM cells in vivo, we established a MM xenograft model by transplanting ILF2 ASO−resistant GFP⁺Luc⁺ JJN3 cells into NSG mice. The mice were randomized based on their bioluminescence-based tumor burden and then treated for 1 week with NT or ILF2 ASOs in the presence or absence of NSC (Supplementary Fig. 3J). Consistent with our hypothesis, the mice that received ILF2 ASOs in combination with NSC had a significantly lower tumor burden than those that received NT ASOs in combination with NSC (Fig. 3G).

Together, these data support the hypothesis that DNA2 inhibition plays a role in promoting MM cells' survival in the context of DNA damage activation-induced metabolic reprogramming, such as that induced by ILF2 depletion.

## DNA2 is essential for lowering mitochondrial ROS production in MM cells

To dissect the mechanistic basis of DNA2 inhibition−induced synthetic lethality in the context of ILF2 depletion, we evaluated whether DNA2 activity is essential to maintaining OXPHOS, upon which ILF2 ASO−resistant cells rely to survive. To this end, we analyzed mitochondrial respiratory activity in NT ASO− and ILF2 ASO−treated JJN3 cells exposed to NSC for 3 days (Fig. 4A and Supplementary Fig. 4A). Compared with NT ASO−treated cells exposed to NSC, ILF2 ASO−treated cells exposed to the DNA2 inhibitor had significantly decreased maximal OCRs and NAD/NADH levels (Supplementary Fig. 4B) and higher mitochondrial ROS production (Fig. 4B).

Mitochondrial DNA (mtDNA) is arranged and packaged in mitochondrial nucleoids which are close to mitochondrial cristae[20], the primary site of the OXPHOS machinery[21]. The mitochondrial cristae and mtDNA interact to maintain mitochondrial integrity[22]. Germline

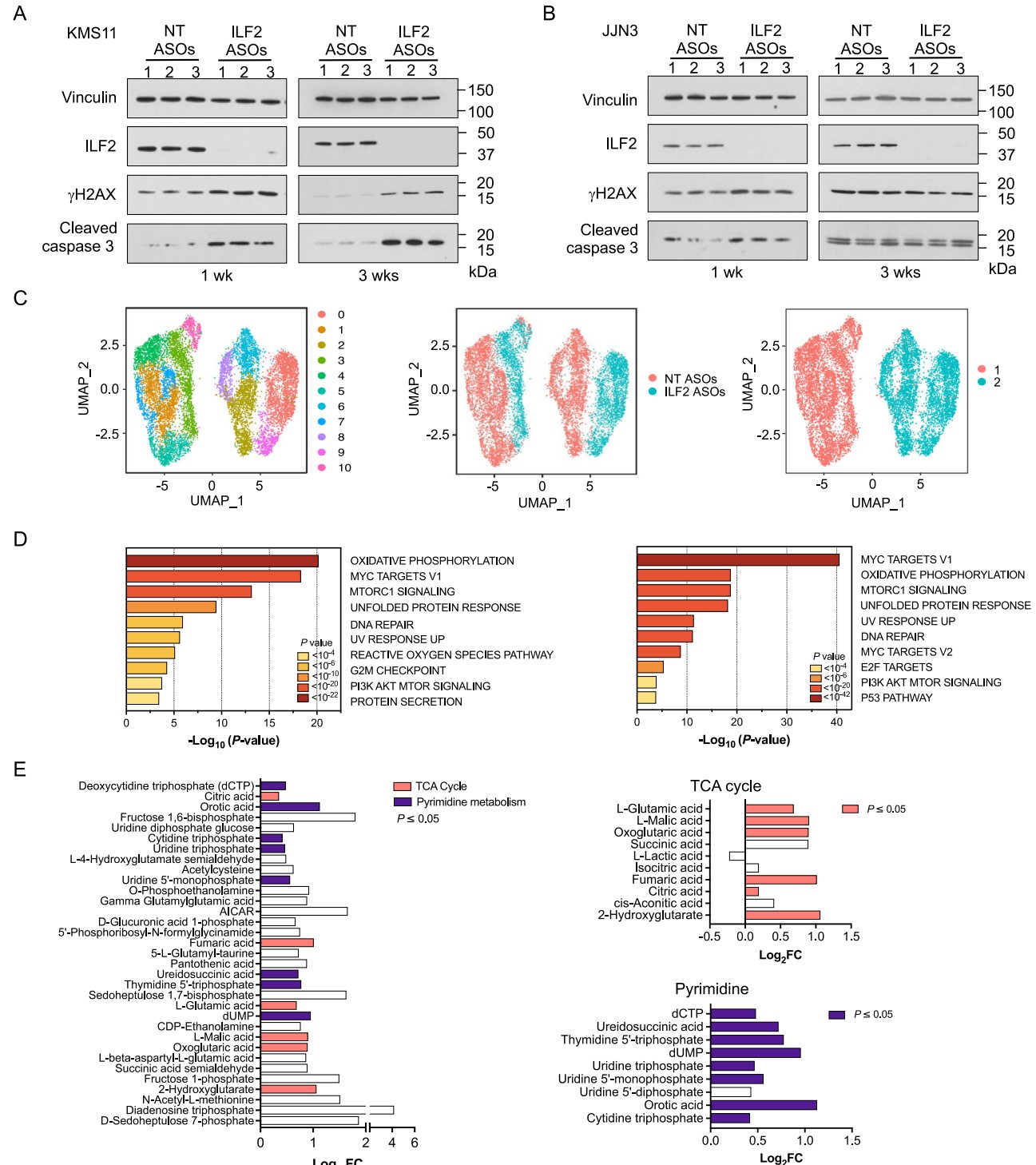

**Fig. 2 | Metabolic reprogramming mediates MM cells' resistance to DNA damage activation. A** Western blot analysis of ILF2, γH2AX, and cleaved caspase 3 in KMS11 cells treated with NT or ILF2 ASOs (0.5 μM) for 1 week (wk; left) or 3 weeks (right). Vinculin was used as a loading control. Every experiment was performed in triplicate (1–3). **B** Western blot analysis of ILF2, γH2AX, and cleaved caspase 3 in JJN3 cells treated with NT or ILF2 ASOs (1 μM) for 1 week (wk; left) or 3 weeks (right). Vinculin was used as a loading control. Every experiment was performed in triplicate (1–3). **C** Uniform manifold approximation and projection (UMAP) of scRNA-seq data displaying pooled (*n* = 2 independent experiments) single JJN3 cells after 3 weeks of NT ASO (*n* = 7041 cells) or ILF2 ASO (*n* = 4462 cells) treatment. Different colors represent the cluster (left), sample origin (middle) and the 2 identities of the main clusters (right). Cluster 10 which included basal apoptotic cells was removed

from the pathway enrichment analysis shown in Fig. 2D. **D** Pathway enrichment analysis of the significantly upregulated genes in ILF2 ASO–treated cells compared with NT ASO–treated cells in the major clusters 1 (left) and 2 (right) shown in Fig. 2C (adjusted *P* ≤ 0.05). The top 10 Reactome gene sets are shown. **E** Log₂ fold change (FC) of all significant metabolites that were significantly enriched in JJN3 cells treated with ILF2 ASOs for 3 weeks compared with cells treated with NT ASOs (left). The significant metabolites in the tricarboxylic acid cycle pathway (top right, *P* = 0.016), and the pyrimidine pathway (bottom right, *P* < 0.001) are highlighted in pink and violet, respectively (right) (*n* = 2 independent replicates per group; adjusted *P* ≤ 0.05). A detailed description of the statistical analysis is included in "Methods" section. Source data are provided as a Source Data file.

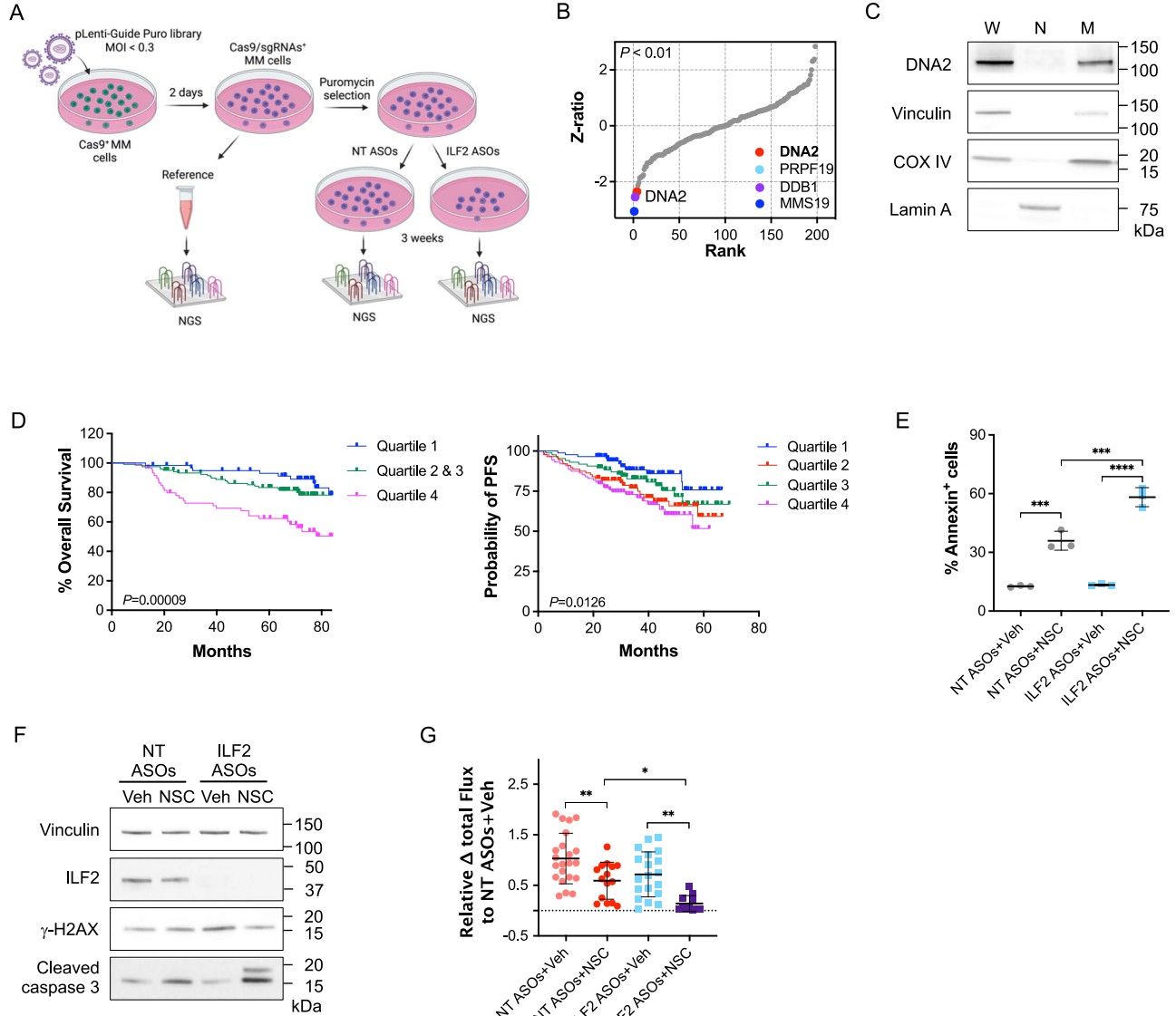

**Fig. 3 | DNA2 is essential for maintaining MM cells' survival after DNA damage–induced metabolic reprogramming. A** Schematic of the CRISPR/Cas9 screening. Stable Cas9+ JJN3 or Cas9+ KMS11 cells were transduced with a library of pooled sgRNAs targeting 196 genes involved in several DNA repair pathways. A portion of cells was collected as a reference sample after 48 h of transduction. Cells were continuously cultured under puromycin selection and treated with NT or ILF2 ASOs for 3 weeks. ILF2 sensitizer genes were identified using deep sequencing of the sgRNA barcodes and the drugZ algorithm to assess differences in the representation of all sgRNAs between NT ASO− and ILF2 ASO−treated cells across the 3 independent sets of experiments. MOI, multiplicity of infection; NGS, next-generation sequencing. **B** Ranking of the DNA repair genes whose sgRNAs were significantly depleted in ILF2 ASO−treated JJN3 cells as compared with NT ASO−treated cells. The inset shows genes on the top ranks (adjusted $P < 0.01$); DNA2; $P = 0.00931$. **C** Western blot analysis of DNA2 in whole-cell lysates (W), nuclei (N), and mitochondria (M) isolated from JJN3 cells. Vinculin, Lamin A, and COX IV were used as the loading controls for W, N, and M, respectively. Two biological replicates were performed. **D** Left, Kaplan–Meier plots for overall survival according to *DNA2* expression in MM PCs as evaluated by RNA-Seq analysis. Shown are the median overall survival durations of patients who were enrolled in clinical trials of velcade in combination with revlimid and dexamethasone followed by autologous transplantation ($n = 41$; log-rank $P = 8.758 \times 10^{-5}$). Right, Kaplan–Meier plots of progression-free survival (PFS) according to *DNA2* expression in MM PCs as evaluated by microarray analysis. Shown are the median PFS

durations of patients who were enrolled in the Arkansas Total Therapy 2 and 3 trials and received high-dose chemotherapy followed by autologous transplantation ($n = 351$; $P = 0.0126$). A detailed description of the statistical analysis is included in "Methods" section. **E** Frequencies of apoptotic (annexin V-positive) JJN3 cells after 3 weeks of exposure to NT or ILF2 ASOs (1 μM) followed by 48 h of treatment with vehicle (Veh) or 2 μM NSC. Data are expressed as the mean ± S.D. from one representative experiment performed in triplicate. Statistically significant differences were detected using two-way ANOVA (****$P < 0.0001$; ***$P < 0.001$; NT ASOs +Veh vs NT ASOs+NSC: $P = 0.0002$; NT ASOs+NSC vs ILF2 ASOs+NSC: $P = 0.0003$; ILF2 ASOs+Veh vs ILF2 ASOs+NSC: $P < 0.0001$). **F** Western blot analysis of ILF2, γH2AX, and cleaved caspase 3 in JJN3 cells treated with NT or ILF2 ASOs (1 μM) for 3 weeks prior to receiving NT or ILF2 ASOs alone (Veh) or in combination with 1 μM NSC for 48 h. Vinculin was used as a loading control. **G** Differences in the luciferase signal in NSG mice engrafted with ILF2 ASO−resistant GFP+Luc+ JJN3 cells after receiving NT or ILF2 ASOs alone (NT or ILF2+Veh) or in combination with NSC every day for 7 days. Data are expressed as the mean bioluminescence activity relative to that of the NT ASOs+Veh group [Δ flux of luciferase signal (photons/second, p/s] ± S.D. For each mouse (NT ASOs+Veh, $n = 22$; NT ASOs+NSC, $n = 15$; ILF2 ASOs +Veh, $n = 19$; ILF2 ASOs+NSC, $n = 11$; $n = 3$ independent experiments). Statistically significant differences were detected using one-way ANOVA (**$P < 0.01$; *$P < 0.05$; NT ASOs+Veh vs NT ASOs+NSC: $P = 0.0098$; NT ASOs+NSC vs ILF2 ASOs+NSC: $P = 0.0328$; ILF2 ASOs+Veh vs ILF2 ASOs+NSC: $P = 0.0021$). Source data are provided as a Source Data file.

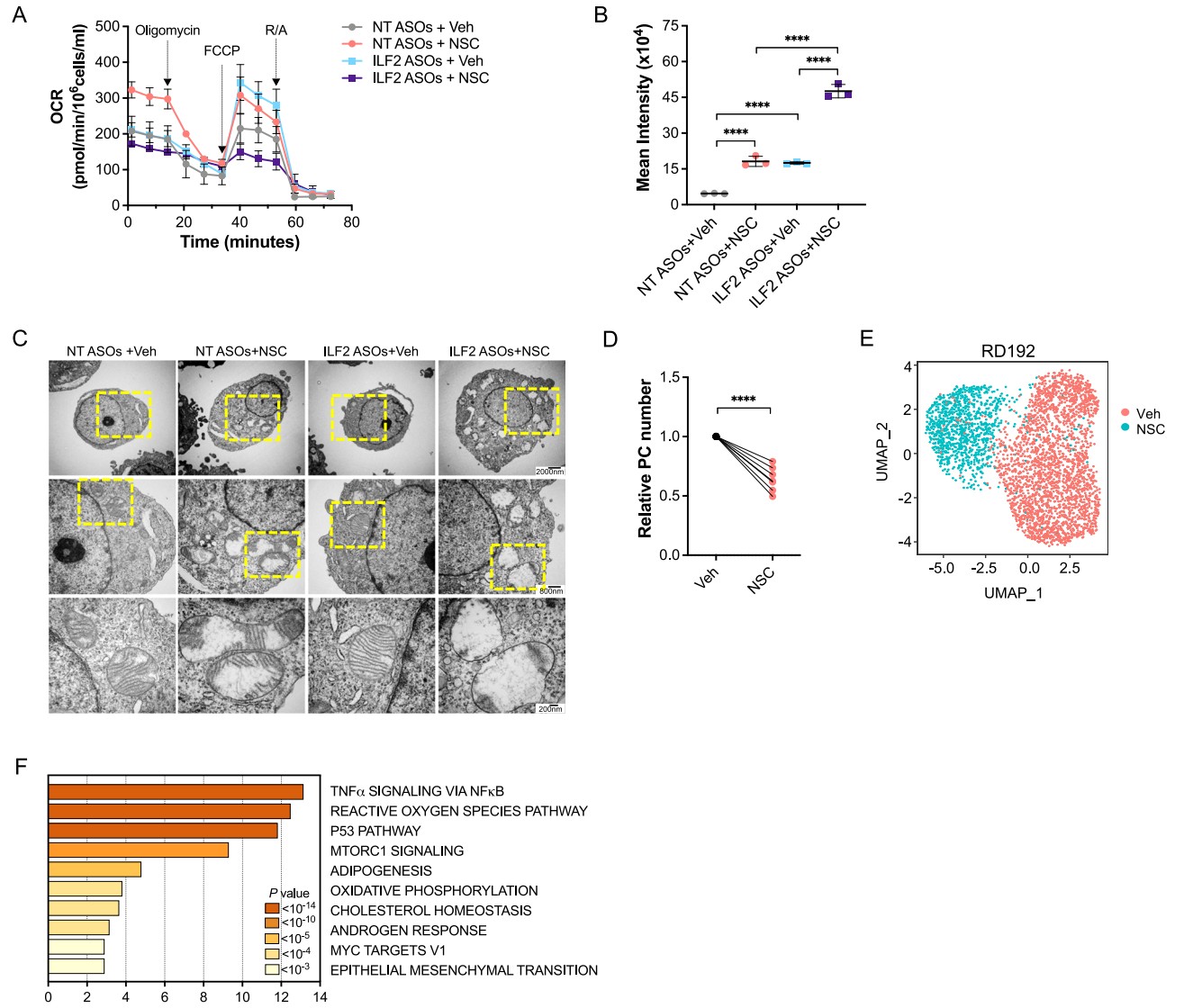

**Fig. 4 | DNA2 is essential for activated OXPHOS in MM cells. A** Oxygen consumption rates (OCRs) in JJN3 cells treated with NT or ILF2 ASOs (1 μM) for 3 weeks prior to receiving ASOs alone or in combination with 1 μM NSC for 72 h. Each data point is the mean ± S.D. of replicates (NT ASOs+Veh, $n = 5$; NT ASOs+NSC, $n = 5$; ILF2 ASOs+Veh, $n = 4$; ILF2 ASOs+NSC, $n = 5$). FCCP, carbonyl cyanide-p-tri-fluoromethoxy-phenylhydrazone; R/A, rotenone/antimycin; Veh, vehicle. Data are expressed as the mean ± S.D. from one representative experiment. Experiments were performed in biological duplicates. **B** ROS production in JJN3 cells treated with NT or ILF2 ASOs (1 μM) for 3 weeks prior to receiving 1 μM NSC for 48 h. Data are expressed as the mean ± S.D. from one representative experiment performed in triplicate. Statistically significant differences were detected using two-way ANOVA (****$P < 0.0001$; NT ASOs+Veh vs NT ASOs+NSC: $P < 0.0001$; NT ASOs+Veh vs ILF2 ASOs+Veh: $P < 0.0001$; NT ASOs+NSC vs ILF2 ASOs+NSC: $P < 0.0001$; ILF2 ASOs +Veh vs ILF2 ASOs+NSC: $P < 0.0001$). **C** Representative transmission electron micrographs showing the mitochondrial ultrastructure of JJN3 cells treated with NT or ILF2 ASOs (1 μM) for 3 weeks prior to receiving 1 μM NSC for 48 h. Scale bars: 7500X, 2000 nm (top); 20,000X, 800 nm (middle); 50,000X, 200 nm (bottom). **D** Numbers of live PCs isolated from the BM of MM patients with PI-based therapy failure ($n = 7$) after treatment with vehicle (Veh) or 2 μM NSC for 48 h over a layer of mesenchymal cells. Data were normalized to each sample's vehicle (Veh)-treated control. Statistical significance was calculated using a paired 2-tailed Student $t$ test (****$P < 0.0001$). **E** UMAP of scRNA-seq data displaying PCs from one MM patient (RD192) with 1q21 amplification, whose disease failed PI-based therapy. Cells were treated for 48 h with vehicle (Veh) or 2 μM NSC over a layer of mesenchymal cells. Different colors represent the sample origins. **F** Pathway enrichment analysis of genes that were significantly upregulated in all 3 NSC-treated MM PC samples shown in Fig. 4E, and Supplementary Fig. 4I, J compared with those treated with vehicle (Veh). The top 10 Hallmark gene sets are shown. Source data are provided as a Source Data file.

*DNA2* loss-of-function mutations induce disruptions in cristae structures. These alterations only affect cells with high metabolic demand and result in early onset myopathies[23,24].

To evaluate whether DNA2 activity inhibition leads to cristae structure perturbations in MM cells, we performed transmission electron microscopy analysis of NT or ILF2 ASO–treated cells exposed to NSC. Although both NT and ILF2 ASO–treated cells exposed to NSC had fragmented mitochondrial cristae structures (Fig. 4C), only ILF2 ASO–treated JJN3 cells exposed to NSC had upregulated expression of

genes involved in respiratory electron transport and ATP synthesis, as an attempt to compensate for the decline in mitochondrial activity and maintain their survival (Supplementary Fig. 4C–E).

Together, these data suggest that MM cells with higher mitochondrial respiration demand rely on repairing mitochondrial DNA damage-induced by increased ROS production and thus have enhanced sensitivity to the inhibition of DNA2, which leads to these cells' apoptosis by inducing mitochondrial cristae structure perturbations.

Given that previous studies in cell lines, mouse xenografts and patient-derived tumor samples demonstrated that a shift from glycolysis to high mitochondrial energy metabolism is sufficient to promote PI resistance[25], and that higher levels of *DNA2* expression were associated with worse progression-free survival after PI-based therapy (Supplementary Fig. 3G), we evaluated whether DNA2 activity inhibition was synthetically lethal in plasma cells (PCs) isolated from patients whose disease failed PI-based therapy. Two days of NSC treatment at a dose that did not deplete PCs isolated from healthy donor BM (Supplementary Fig. 4F) in co-culture with mesenchymal cells significantly reduced NAD/NADH levels (Supplementary Fig. 4G), increased mitochondrial ROS production (Supplementary Fig. 4H), and led to cell death (Fig. 4D) in PCs isolated from patients whose disease failed PI-based therapy (Supplementary Table 2). scRNA-seq analysis of PCs isolated from the co-cultures (Fig. 4E, and Supplementary Fig. 4I, J) showed that NSC–treated PCs had a significant increase in the expression of genes involved in the ROS and respiratory electron transport pathways (Fig. 4F), which is consistent with the results observed in NSC–treated JJN3 cells. These data suggest that DNA2 is essential to counteracting oxidative DNA damage and maintaining mitochondrial respiration in the context of metabolic reprogramming.

## Discussion

We developed ILF2 ASOs to induce DNA damage in 1q21 MM cells and to assess whether 1q21 MM cells become resistant to persistent DNA damage activation-induced by impaired DNA repair pathways. Consistent with longstanding clinical data[12], our findings demonstrate that 1q21 MM cells can eventually overcome the deleterious effects of DNA damage, which confirms that DNA damage resistance is a major barrier to effective DNA-damaging anticancer therapy in MM. Using multiple unbiased analyses, we found that DNA damage–resistant MM cells rely on mitochondrial metabolism to maintain survival and we identified DNA2 as an essential effector of MM cells' resistance to agents that induce metabolic adaptation (Supplementary Fig. 4K).

Previous studies investigating the role of DNA2 in cancer pathogenesis and progression showed that DNA2 overexpression supports breast and pancreatic cancer cell survival by overcoming chemotherapy- or radiotherapy-induced replication stress at the DNA replication fork[26,27]. Our functional data revealed a different role of DNA2 in cancer cells and demonstrated that DNA2 is essential to maintaining MM cells' survival under DNA damage-induced metabolic reprogramming. Indeed, *DNA2* expression levels were highly correlated with poor prognosis after melphalan- or PI-based therapy, which supports the hypothesis that DNA2 activity inhibition represents a synthetically lethal approach to targeting MM cells with high mitochondrial demand. Interestingly, although DNA2 expression levels were significantly higher in patients with 1q21 gain/amplification, DNA2 activity inhibition significantly depleted both 1q21 and non-1q21 amplified PCs from patients that were refractory to PI-based therapy, possibly because other mechanisms independent of 1q21 (i.e., metabolic rewiring) account for disease progression at the time of MM relapse after PI-based therapy. These data suggest that DNA2 inhibition may have therapeutic potential for MMs that rely on OXPHOS to maintain survival independently of the genetic alterations. Together, these data support the development of selective small molecules targeting mitochondrial DNA2's nuclease activity in vivo to affect metabolic rewiring in cancer cells after therapy resistance.

Consistent with our findings, other studies showed that DNA2 plays a role in maintaining mitochondrial functional integrity. Loss-of-function germline mutations in *DNA2* cause cells to accumulate mitochondrial DNA damage and can lead to various mitochondrial diseases affecting energy metabolism in human organs and tissues that rely on OXPHOS to function[23,28]. While these findings support the role of DNA2 in maintaining mitochondrial homeostasis, they also suggest that targeting DNA2 can lead to widespread toxicity in normal tissues. However, mice heterozygous for *DNA2* loss-of function mutations are viable, which suggests that there is a therapeutic window to inhibit DNA2 activity in the context of cancers with DNA2 overexpression, such as MM that has relapsed after PI-based treatment[27].

In conclusion, our study revealed a unique vulnerability of MM cells that are forced to use oxidative phosphorylation to overcome DNA damage activation. Given that metabolic reprogramming is a hallmark of cancer progression, further studies will clarify whether therapeutically targeting DNA2 has a broad spectrum of anti-cancer applications.

## Methods

The research complies with all relevant ethical regulations: MD Anderson Cancer Center IRB-approved human sample protocol PA15-0926; MD Anderson Cancer Center IACUC-approved mouse protocol 0000931-RN03.

### MM cell lines and primary MM samples

JJN3 cells were obtained from DSMZ (ACC 541). KMS11 cells were obtained from JCRB Cell Bank (JCRB1179). MM1R (CRL-2975), H929 (CRL-3580) and RPMI-8226 (CCL-155) cells were obtained from ATCC. Human bone marrow-derived mesenchymal stem cells (hMSC) were kindly provided by Dr. Michael Andreeff. Mycoplasma testing was routinely performed on all cell lines, and cell identity was validated by STR DNA fingerprinting using the Promega 16 High Sensitivity STR Kit. Primary BM samples from patients with MM relapsed disease after PI-based therapy and referred to the Department of Lymphoma and Myeloma at MD Anderson Cancer Center or the Department of Medicine and Surgery at the University of Parma were obtained after written informed consent with the approval of the institutions' respective Institutional Review Boards (IRBs) and in accordance with the Declaration of Helsinki. The consent to publish the exact age and sex of the patients included in the study was obtained by the Department of Lymphoma and Myeloma at MD Anderson Cancer Center or the Department of Medicine and Surgery at the University of Parma. Patient characteristics are included in Supplementary Table 2. BM samples from healthy donors were obtained from AllCells.

The authors acknowledge the limitation of using MM cell lines to perform the experiments described in the manuscript.

### Cell culture and viability assays

MM cell lines (KMS11, JJN3, RPMI-8226, H929, and MM1R) were cultured in RPMI 1640 medium supplemented with 10% fetal bovine serum, 1% penicillin/streptomycin, and 0.1% amphotericin B (all from Gibco). Human MSCs were cultured in MEM alpha, GlutaMAX medium supplemented with 10% MSC FBS, 1% penicillin/streptomycin, and 0.1% amphotericin B (all from Gibco). Cell cultures were maintained at 37 °C in 5% $CO_2$. Cells were constantly seeded at a density of 200,000 cells/mL independently of the type of treatment they received. Total cell viability was evaluated using trypan blue staining.

Primary BM mononuclear cells isolated from MM patients or healthy donors were enriched in CD138$^+$ PCs using magnetic sorting with the CD138 Microbead Kit (Miltenyi Biotec). Cells were plated in 48-well plates previously seeded on a layer of human BM-derived mesenchymal cells.

### Drug treatments

ASOs were designed and synthesized by IONIS Pharmaceuticals under a collaborative agreement. The list of mouse and human ILF2 ASOs used in this study are included in Supplementary Table 1. NT and ILF2 ASOs were prepared in culture medium supplemented with 10% fetal bovine serum to achieve the indicated concentrations. ASOs were delivered to the cells by free uptake. For in vitro single-agent assays, KMS11, JJN3, MM1R, H929, and RPMI-8226 cells were initially treated with 0.1, 0.5, 1, 2, or 2.5 μM ASOs for 7 days. ASOs were added every

2 days together with fresh media. For combination therapy studies, the cells were treated with melphalan (Sigma), bortezomib (Tocris), brequinar (Sigma), IACS-010759 (IACS), NSC105808 (Chemspace), or C5 (AOB9082, Aobious, Inc.) at the concentrations and times indicated in the figure legends in the presence or absence of NT or ILF2 ASOs (KMS11, 0.5 µM; JJN3, 1 µM; RPMI-8226, 1 µM; MM1R, 1 µM; H929, 2 µM).

Primary PCs isolated from MM patients and healthy donors were treated with NSC105808 at the concentrations and times indicated in the figure legends prior to being analyzed.

## Mouse experiments

Animal experiments were approved by MD Anderson's Institutional Animal Care and Use Committee and performed in accordance with the Animal Welfare Act in accordance with the MD Anderson Cancer Center IACUC-approved mouse protocol 0000931-RN03.

For in vivo tolerability studies in an immune-competent mouse strain, BALB/cJ mice (strain #000651) were obtained from Jackson Laboratory. Mice were treated with PBS or ASOs targeting murine *Ilf2* or human *ILF2* at a dose of 50 mg/kg delivered twice weekly by intraperitoneal injection for 4 weeks. At the end of the study, peripheral blood samples were collected for blood chemistry evaluation. Mice were euthanized and the liver, kidneys, and lungs from each mouse were weighed and collected for *Ilf2* expression quantification. *Ilf2* expression was only quantified in the kidneys and lungs of the mice because liver cells do not express *Ilf2*.

For xenograft experiments, 4-week-old NSG mice (strain #005557) were obtained from the Jackson Laboratory and maintained in a pathogen-free environment at MD Anderson and housed in a barrier facility at 25° C under ambient oxygen conditions in a 12-h light/12-h dark cycle under 50% humidity. Mice were monitored daily and humanely euthanized at the first sign of morbidity. Moribund animals (animals that were not eating, drinking, or eliminating, that were exhibiting cachexia, and/or that had inhibited mobility), and animals showing obvious signs of stress were euthanized in accordance with the MD Anderson Cancer Center IACUC-approved mouse protocol 0000931-RN03.

Mice used for the experiments were all females. Female recipient mice enable better engraftment of tumoral cells[29].

Mice were maintained under specific-pathogen-free conditions. NSG mice were sublethally irradiated prior to receiving GFP+Luc+ KMS11 cells ($2 \times 10^6$) or ILF2 ASO-resistant GFP+Luc+ JJN3 cells ($1 \times 10^6$) via tail vein injection. Mice harboring GFP+Luc+ KMS11 cells were injected with luciferin and anaesthetized, and their tumor burden was determined by live luminosity using the IVIS Spectrum bioluminescence imaging system (PerkinElmer). Mice were randomized based on the level of tumor burden detected by bioluminescence imaging (total flux; proton/sec) at day 0 (before any treatment). Randomized mice were assessed for tumor burden after 7 doses of ASOs (50 mg/kg) and after another 3 doses of ASOs (25 mg/kg) in combination with melphalan (2.5 mg/kg). Moribund mice were humanely euthanized, and target engagement was evaluated by real-time PCR in sorted GFP+ KMS11 cells. Mice harboring ILF2 ASO-resistant GFP+Luc+ JJN3 cells were randomized based on the level of tumor burden detected by bioluminescence imaging before receiving NT or ILF2 ASOs (25 mg/kg) alone or in combination with IACS-010759 (10 mg/kg) or NSC (10 mg/kg) in independent experiments. Survival curves were analyzed using the Mantel−Cox log-rank test.

## Flow cytometry analysis and Fluorescence-activated Cell Sorting (FACS)

Flow cytometry and FACS experiments were performed using the BD LSR Fortessa and BD Influx Cell Sorter (BD Biosciences), respectively. The FlowJo software (https://www.flowjo.com) was used to analyze the data. All experiments included single-stained controls and were performed at the South Campus Flow Cytometry & Cellular Imaging

Facility at MD Anderson Cancer Center. These following antibodies were used for quantitative flow cytometry and FACS analyses: anti-human CD138 (BioLegend, #347207, dilution 1:20); anti-mouse CD45 (BioLegend, #103116, dilution 1:20), and anti-human CD90 (BioLegend, #328118, dilution 1:20). Cell viability was assessed by DAPI staining (ThemoFisher Scientific, #62248, dilution 1:5000). The cell surface marker expression panel and the gating strategies used for the identification, quantification, and purification of MM cells by flow cytometry are described in Supplementary Table 3.

## Apoptosis assays

KMS11, JJN3, MM1R, H929, and RPMI-8226 cells were treated with NT or ILF2 ASOs for 1 or 3 weeks prior to receiving either ASOs alone or ASOs in combination with melphalan, bortezomib, IACS-010759, brequinar, or NSC at the concentrations and times specified in the figure legends. The frequencies of apoptotic cells were determined by flow cytometry using the annexin-V assay (BD Bioscience, #88-8007-74).

## Mitochondrial ROS production

JJN3 cells were treated with 1 µM NT or ILF2 ASOs prior to receiving 1 µM IACS-010759 or 1 µM NSC for 48 h. PCs were treated with vehicle or 2 µM NSC for 48 h. Mitochondrial ROS production was quantified using the MitoSOX Red assay (Invitrogen, M36008) following the manufacturer's protocol.

## NAD/NADH quantification

JJN3 cells were treated with 1 µM NT or ILF2 ASOs prior to receiving 1 µM NSC for 48 h. PCs were treated with vehicle or 2 µM NSC for 48 h. Intracellular levels of NAD/NADH were measured by plate luminescence detection using the NAD/NADH-GloTM quantitation kit (Promega, G9071) according to the manufacturer's instructions. Luminescence levels in relative light units were measured using a Victor X2 multimode microplate reader (PerkinElmer) and normalized to the total cell number.

## Western blot analysis

Cell pellets were harvested and resuspended in Mammalian Cell & Tissue Extraction Kit buffer (BioVision Incorporated, K269) and incubated for 10 min on ice. Protein lysates were collected after centrifugation at $14,000 \times g$ for 20 min at 4 °C. The total amount of protein was quantified using the Qubit Protein Assay Kit and a Qubit Fluorometer (Thermo Fisher). Sodium dodecyl sulfate–polyacrylamide gel electrophoresis and Western blotting were performed using pre-cast NuPAGE Bis-Tris 4–12% mini-gels (Invitrogen) with 1X MOPS buffer (Invitrogen), following the manufacturer's instructions. The primary antibodies anti-ILF2/NF45 (Santa Cruz, sc365068, dilution 1:500), anti-vinculin (Sigma, V9131, dilution 1:2000), anti-γH2AX (Cell Signaling, 2577 S, dilution 1:500), anti-cleaved caspase 3 (Cell Signaling, 9661 S, dilution 1:500), and anti-Cas9 (Cell Signaling, 14697 S, dilution 1:1000), in addition to secondary anti-mouse and anti-rabbit digital antibodies (Kindle Biosciences LLP, dilution 1:2000), were used. Membranes were developed using SuperSignal West Pico PLUS Chemiluminescent Substrate (Thermo Fisher) and imaged using a KwikQuant Imager and software (Kindle Biosciences LLP).

## Quantitative real-time PCR

In xenograft experiments, RNA was extracted from sorted GFP+ KMS11 cells using the Arcturus PicoPure RNA isolation kit (Applied Biosystems), and cDNA was synthesized using Arcturus RiboAmp HS PLUS RNA Amplification Reagents (Applied Biosystems) according to the manufacturer's protocol. Real-time PCR was performed using the TaqMan Universal PCR Master Mix (Applied Biosystems) and a 7500 Real-Time PCR System (Applied Biosystems). Each condition was performed in duplicate. *ACTIN* was used as a housekeeping gene. The expression level of *ILF2* was normalized to that of *ACTIN*.

**Histological analyses.** Formalin-fixed paraffin-embedded mouse BM or liver sections were prepared for antibody detection and hematoxylin and eosin staining according to standard procedures. IHC was performed at the Dana Farber/Harvard Cancer Center Specialized Histopathology Core (Boston, MA). Samples were stained with anti-human ILF2 (H-4, Santa Cruz, dilution 1:100), and anti-human cleaved caspase 3 (D3E9, Cell Signaling, dilution 1:100).

**CRISPR/Cas9 library screening of sgRNAs targeting DNA repair genes.** The CRISPR/Cas9 library of pooled sgRNAs targeted 196 genes involved in DNA repair pathways and the DNA damage response regulation was designed at Cellecta using a proprietary algorithm with a coverage of 10 sgRNAs/gene (Supplementary Data 1). The library was cloned into the pLentiGuide-Puro lentiviral vector. KMS11 or JJN3 cells were transduced with the pCW-Cas9-Blast vector (#83481, Addgene) to establish stable Cas9+ KMS11 and Cas9+ JJN3 cells. Cas9+ cells were selected with 5 μg/mL blasticidin. Cas9+ cells were infected with a library of pooled sgRNAs targeting DNA repair pathways at a multiplicity of infection of <0.3 at 1000x coverage, and $8 \times 10^6$ cells were collected at 48 h after the transduction and used as a reference sample. Cells were selected with 1 μg/mL puromycin and continuously treated with NT or ILF2 ASOs (0.5 μM) for 3 weeks before collection. Cells were pelleted and frozen at −80 °C before further processing for DNA extraction. Every experiment was independently repeated 3 times. DNA was extracted with DNeasy Blood & Tissue Kits (Qiagen) according to the manufacturer's protocol. Genomic DNA was used for the PCR template using a mixture of 8 staggered primers with NEBNext Q5 Hot Start HiFi PCR Master Mix with an initial denaturing at 98 °C for 1 min, denaturing at 98 °C for 10 s, annealing at 64 °C for 20 s, elongation at 72 °C for 30 s, and final elongation for 2 min. PCR cycles for each sample were controlled to the minimal levels at which the target bands could be seen in 2% agarose TAE gel to ensure unbiased PCR amplification. Each sample had a different reverse primer that differed in only an 8-digit barcode. The pooled Illumina library was then subjected to NextSeq550 high-output sequencing with >1000x coverage per sample. For data analysis, raw reads were demultiplexed without any tolerance of barcode and then mapped using Bowtie with a single-base mismatch tolerance. Read counts for each sgRNA were enumerated. For the identification of genes sensitizing cells to ILF2 ASOs treatment, the reads were normalized, and the abundance difference between the NT ASO–sensitive and ILF2 ASO–sensitive cells for each sgRNA were calculated and corrected for multiple hypothesis testing using the drugZ algorithm[16].

**Bulk RNA-seq analysis**
RNA was extracted from KMS11 or JJN3 cells treated with NT or ILF2 ASOs using the RNeasy kit (Qiagen). Estimates of gene expression were generated by pseudo-aligning FASTQ files against human genome GRCh38.p12 (Ensembl version 94) using Kallisto with the default options[30,31]. Differential expression analysis was conducted using DESeq2 in R version 3.5.1[32]. Separate differential expression analyses were conducted to compare time points or treatments within each cell line. In addition, a multivariate analysis was performed which that included the time point, the treatment, and an interaction term to estimate treatment-induced differences in gene expression changes over time. Biologically relevant gene sets containing multiple differentially expressed genes were identified by analyzing the results of differential expression analyses using GSEA-pre-ranked analysis, as implemented in the FGSEA package[33].

**scRNA-seq analysis**
JJN3 cells were treated with 1 μM NT or ILF2 ASOs for 3 weeks. In parallel experiments, JJN3 cells exposed to 1 μM NT or ILF2 ASOs for 3 weeks were treated with vehicle or 1 μM NSC for 48 h. Primary PCs were treated with 2 μM NSC for 48 h. Live cells were sorted by flow cytometry and subjected to scRNA-seq analysis. Experiments were performed in biological duplicates. Sample preparation and sequencing were performed at The University of Texas MD Anderson Cancer Center's Sequencing and Microarray Facility. Samples were normalized for input onto the Chromium Single Cell A Chip Kit (10x Genomics), in which single cells were lysed and barcoded for reverse-transcription. The pooled single-stranded, barcoded cDNA was amplified and fragmented for library preparation. During library preparation, appropriate sequence primer sites and adapters were added for sequencing on a NextSeq 500 sequencer (Illumina). After sequencing, FASTQ files were generated using the cellranger mkfastq pipeline (version 3.0.2). The raw reads were mapped to the human reference genome (refdata-cellranger-GRCh38-3.0.0) using the cellranger count pipeline. The digital expression matrix was extracted from the filtered_feature_bc_matrix folder outputted by the cellranger count pipeline. Multiple samples were aggregated using the cellranger aggr pipeline. The digital expression matrix was analyzed with the R package Seurat (version 3.0.2) to identify different cell types and signature genes for each. Cells with fewer than 500 unique molecular identifiers or greater than 50% mitochondrial expression were removed from further analysis. The Seurat function NormalizeData was used to normalize the raw counts. Variable genes were identified using the FindVariableFeatures function. The ScaleData function was used to scale and center expression values in the dataset, and the number of unique molecular identifiers was regressed against each gene. Uniform manifold approximation and projection was used to reduce the dimensions of the data and the first 2 dimensions were used in the plots. The FindClusters function was used to cluster the cells. Marker genes for each cluster were identified using the FindAllMarkers function.

**Targeted metabolomic analysis**
JJN3 cells were pre-incubated with 1 μM NT or ILF2 ASOs for 3 weeks prior to receiving 1 μM NSC for 48 h. Live cells ($1 \times 10^6$) were sorted by flow cytometry and subjected to metabolomic analysis. Metabolites were extracted from dry cell pellets using 1 mL of ice-cold 0.1% ammonium hydroxide in 80/20 (v/v) methanol/water. Extracts were centrifuged at $17,000 \times g$ for 5 min at 4 °C, and supernatants were transferred to clean tubes and evaporated to dryness under nitrogen. Dried extracts were reconstituted in deionized water and 10 μL were injected for analysis by ion chromatography–mass spectrometry (IC-MS). For mobile phase A, water was chosen, and for mobile phase B (MPB), water containing 100 mM potassium hydroxide was chosen. The Thermo Scientific Dionex ICS 5000+ system, which included a Thermo IonPac AS11 column (4-μm particle size, 250 × 2 mm) with the column compartment kept at 30 °C, was used to perform IC-MS with a total run time was 50 min. Methanol was delivered by an external pump and combined with the eluent via a low dead volume mixing tee. Data were acquired using a Thermo Orbitrap Fusion Tribrid Mass Spectrometer under ESI negative ionization mode at a resolution of 240,000. Raw data files were imported to Thermo Trace Finder software for final analysis. The relative abundance of each metabolite was normalized by each sample's live cell count.

The analyses were performed using R version 3.6.3 (2020-02-29). Limma analysis was used to access differential regulation of metabolites between ILF2 ASOs and NT ASOs (2 samples in each treatment). We extracted log2 fold change (logFC) and $p$ values from the linear regression model fitting. Benjamini-Hochberg (BH) method was applied for a multiplicity adjustment. We classified each metabolite based on the adjusted $p$ value and logFC into one of the three groups, upregulated ($p < 0.05$ and logFC > 0), downregulated ($p < 0.05$ and logFC < 0), and unchanged (the rest group including $p >= 0.05$).

## Copy number analysis

Total DNA from JJN3 cells, was extracted using the DNeasy Blood & Tissue Kit (Qiagen, Valencia, CA). Single nucleotide polymorphism (SNP) analysis was performed in NT and ILF2 ASO–treated JJN3 cells after 3 weeks of ASO exposure using the Human CytoSNP v2.1 Bead-Chip Kit (Illumina).

Allele-Specific Copy Number Analysis of Tumors (ASCAT) was performed to identify allele-specific copy number alterations. The Bioconductor package, ASCAT (version 3.1.1), and the hg38 reference genome were used for the analysis. More specifically, SNP name, chromosome, position, the log R ratios, and B allele frequencies generated by GenomeStudio were put into ASCAT.

## Immunofluorescence microscopy

KMS11 or JJN3 cells were fixed and permeabilized using IntraPrep Permeabilizaton Reagent (Beckman Coulter) following the manufacturer's protocol. Samples were incubated with the primary antibodies anti-γH2AX (Cell Signaling, 2577 S), anti-DNA2 (Invitrogen, PA5-66086), and anti-TOM20 (Santa Cruz, sc17764) at a dilution of 1:200 overnight at 4 °C, washed 3 times with PBS, and then incubated with fluorescently labeled goat anti-rabbit 488 secondary antibody (Invitrogen, 2156517) at a dilution of 1:400 for 1 h at room temperature. Nuclei were stained with 1 µg/mL DAPI at a dilution of 1:1000. Samples were washed 3 times with PBS and coverslips were mounted with Prolong Gold Antifade reagent (Life Technologies). Images were acquired using a confocal microscope (Nikon Instruments Inc.) and analyzed using Image J software v1.51U (https://imagej.nih.gov/ij/) or using a Delta Vision OMX Blaze V4 Super-Resolution System with 62X magnification.

## Transmission electron microscopy

JJN3 cells ($3 \times 10^6$) were washed twice with PBS and fixed in 4% paraformaldehyde solution, pH 7.3. Fixed samples were washed in 0.1 M sodium cacodylate buffer, treated with 0.1% Millipore-filtered cacodylate buffered tannic acid, and postfixed with 1% buffered osmium tetroxide and 1% Millipore-filtered uranyl acetate. Samples were dehydrated using increasing concentrations of ethanol, embedded in LX-112 medium, and polymerized in a 60 °C oven for approximately 3 days. Ultrathin sections were cut in an Ultracut microtome (Leica), stained with uranyl acetate and lead citrate in an EM Stainer (Leica), and examined using a JEM 1010 transmission electron microscope (JEOL) at an accelerating voltage of 80 kV. Digital images were obtained using the Advanced Microscopy Techniques Imaging System (Advanced Microscopy Techniques Corp) using 7500X, 20,000X, and 50,000X magnification.

## Quantification of mitochondrial respiration

OCR was quantified by the Seahorse Mito Stress Test assay (Agilent Technologies). JJN3 cells were treated with 1 µM NT or ILF2 ASOs for 3 weeks prior to receiving 1 µM IACS-010759 or NSC for 72 h. After exposure to IACS-010759 or NSC, cells were washed twice with PBS and resuspended in prewarmed Seahorse basal medium supplemented with 1 mM pyruvate, 2 mM glutamine, and 5 mM glucose, pH 7.4. Cells at a density of $1.5 \times 10^6$ cells/mL were plated in at least 4 replicates on 96-well Seahorse cell culture plates previously coated with Cell-Tak (Corning) according to the manufacturer's instructions. Once plated, the cells were subjected to gentle centrifugation. OCR was determined using the Seahorse XFe96 analyzer according to the manufacturer's instructions. OCR values were obtained at baseline (3 initial measurements) and post-injections of the Seahorse XF Mito Stress Test Kit reagents oligomycin (1.5 µM), carbonyl cyanide-p-trifluoromethoxy-phenylhydrazone (1 µM), and rotenone/antimycin (0.5 µM). All measurements were quantified using the Mito Stress Test Generator and normalized to the number of viable cells.

## Mitochondria and nuclear fractionation

A mitochondria isolation kit (Abcam, ab110171) was used to prepare the large organelles/debris and intact mitochondria fractions from JJN3 cells. Briefly, cell pellets were frozen and thawed to weaken cell membranes. Cell pellets were resuspended in the extraction buffer and homogenized following the manufacturer's procedures. After the last centrifugation step of mitochondrial isolation, the supernatants were collected for further nuclear isolation using the nuclear extraction buffer from a nuclear/cytosol fractionation kit (Biovision; K269) following the manufacturer's procedures. Mitochondrial and nuclear proteins were quantified using the Qubit Protein Assay kit. WB analysis was performed using the following primary antibodies: anti-DNA2 (Invitrogen, PA5-8167, dilution 1:1000), anti-vinculin (Sigma, V9131, dilution 1:2000), anti-COX IV (Cell Signaling, 4850 S, dilution 1:1000), and anti-Lamin A (Abcam, ab26300, dilution 1:1000).

## Clinical correlations

To evaluate whether *DNA2* expression was correlated with poorer progression-free survival (PFS) in MM patients treated with high-dose melphalan, we analyzed the cumulative survival rate of 351 newly diagnosed MM patients enrolled in the Arkansas Total Therapy 2 and 3 trials and treated with high-dose chemotherapy and stem cell transplantation[34] using data deposited in GSE2658. Patients were stratified in 4 quartiles based on *DNA2* expression. The Kaplan-Meier curves were plotted, and the log-rank test was performed to test the difference in survival distributions among the 4 groups.

To evaluate whether *DNA2* expression was correlated with poorer PFS in MM patients treated with PI-based therapy, we used the publicly available IA16 CoMMpass dataset from the Multiple Myeloma Research Foundation. We obtained RNA-seq data from the Salmon V7.2 Filtered Gene TPM file. We used IA16_FlatFile files for demographic, disease, and survival data. *DNA2* gene (ENSG00000138346) expression levels were identified and matched to baseline patient data. Only patients who did not undergo autologous stem cell transplant were included in the survival analysis. Patients were further divided into subgroups based on the use of immunomodulatory agents or PIs during induction therapy. *DNA2* gene expression was analyzed as a continuous variable and further divided into quartiles. All statistical analyses were performed using the BlueSky Statistics 7.40 software package. Normality tests were performed and association testing for categorical variables was done using a chi-squared test. Testing for continuous variables was done with Student *t* test, Mann–Whitney U test, or ANOVA. Progression-free survival was analyzed. Univariate and multivariate Cox proportional hazard models were created to estimate hazard ratios for the association of *DNA2* expression and survival. Multivariate analysis included variables known to be significantly associated with MM outcome. To evaluate whether *DNA2* expression was correlated with overall survival in MM patients treated with a combination of velcade, revlimid and dexamethasone followed by autologous transplantation, we used the publicly available IA16 CoMMpass dataset from the Multiple Myeloma Research Foundation.

Kaplan–Meier curves were constructed for *DNA2* expression quartiles and compared using a log-rank test. A P value of < 0.05 was set for statistical significance.

## Statistics and reproducibility

All statistical data are presented as the mean ± the standard deviation (S.D.) of the mean. The number of replicates in each experiment is indicated in the figure legends. Statistically significant differences were detected using a 2-tailed Student *t* test, one-way ANOVA, or two-way ANOVA as indicated ($****P \leq 0.0001$, $***P \leq 0.001$, $**P \leq 0.01$, $*P < 0.05$). Analyses were performed with the GraphPad Prism 10.1.0 software program (https://www.graphpad.com).

Functional enrichment analysis was performed using the Panther (http://www.pantherdb.org/tools/compareToRefList.jsp) or

the Metascape software[35] packages. The human Hallmark and/or Reactome gene sets were used, and analyses were performed using gene annotation available in 2019-2021. Figure 3A, and Supplementary Figs. 1N, 2L, 3J, 4K were made using Biorender.com. No statistical method was used to predetermine sample size. No data were excluded from the analyses. The mouse experiments were randomized based on the level of tumoral cell engraftment. No statistical method was used to predetermine sample size. The investigators were blinded to allocation during experiments and outcome assessment.

### Reporting summary
Further information on research design is available in the Nature Portfolio Reporting Summary linked to this article.

## Data availability
Data sets generated in this study using RNA-seq and scRNA-seq have been deposited at GEO under accession codes GSE192944 and GSE196766 Source data are provided with this paper.

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

## Acknowledgements
This work was supported by grants from the NCI (R01CA222253 to S.C.) and the Leukemia & Lymphoma Society Multi-Investigator award SCOR-7016-18. S.C. is a Scholar of the Leukemia and Lymphoma Society. N.T. was supported by a Young Investigator Award at the International Myeloma Workshop in 2019 and 2021 and by an ASH Research Restart

Award in 2020. This work used MD Anderson's Advanced Cytometry and Sorting Facility, Advanced Technology Genomics Core Facility, High Resolution Electron Microscopy Facility, Metabolomics Core Facility, and Advanced Microscopy Core Facility, all of which are supported in part by the NIH through the University of Texas MD Anderson Cancer Center Support Grant (P30 CA16672). We also thank Dana-Farber/Harvard Cancer Center for the use of the Specialized Histopathology Core, which provided IHC services and is supported in part by the National Institutes of Health through a Cancer Center Support Grant (5 P30 CA06516). The authors thank Joseph Munch and Helen Chifotides for assistance with manuscript editing.

## Author contributions

S.C. designed and guided the research; N.T., A.S., J.L, N.B., C.J., I.G.-G., V.A., J.J. R-S., P.L., B.W. and A.R. performed experiments; F.M. analyzed scRNA-seq data; Y.Q., M.H. and R.F. performed the statistical analyses; C.C. analyzed the bulk RNA-seq data; L.T. and P.L. performed the metabolomic analyses; V.M., P.S., and D.B.N. processed the primary MM samples included in the studies; C. B-R and R.K-S analyzed the BM and liver biopsies; K. F., K.T., G. T., and G.S. analyzed copy number alterations; M.M., N.G., C.C, M.K., G.G-M., E.M., R.O., A.V. and M.C. made critical intellectual contributions throughout the project; S.C. wrote the manuscript.

## Competing interests

The authors declare no competing interests.
