## [Peer Review File · Nature Communications]

REVIEWER COMMENTS

Reviewer #1 (Remarks to the Author):

Thongon et al. presented an interesting study showing how myeloma cells' metabolic rewiring is an essential mechanism through which the tumor cell can survive DNA damage. Targeting this mechanism suppresses MM cells' ability to survive DNA damage (chemo-induced), and represents a promising new therapeutic target.

Overall the paper is well-written and presents several experiments and data in support of the Authors' narrative and hypothesis. Nevertheless, there are aspects of the paper that require careful revision and the Authors should provide more data to support some of their statements.

Major comments:

The terminology needs to be revised. Amplification 1q is used to define 2 or more extra copies of 1q (Walker et al. Leukemia 2019). These patients are also called double-hit. Authors should carefully distinguish between gain 1q (one extra copy) and amp 1q (>1 extra copy) and investigate if their gene signatures and data are affected by a gene dosage effect (e.g. more 1q copies more effect).

The AST and Liver weight seem to be increased in Figure 1A. Was this significantly higher compared to the controls. Does this data create any conflict with the statement: "We did not observe either consistent histopathological or biochemical ASO-induced alterations, which suggests that Ilf2 depletion does not induce on-target toxicity (Fig. 1A)."

In general, it would be good to add p-values to all boxplots or generate supplementary tables where the p-value of each comparison is reported.

JJN3 is a MAF translocated cell line. It also has 1q gain. Do the Authors think that the sensitivity to ILF2 ASO is due to 1q, or also to MAF? Or in general to high proliferation and aggressive biology?

It is unclear in the narrative why KMS11 was selected. This cell line has 1q, but were there any other reasons?

Figure 1E: was the difference in liver size significant?

Figure 2C: some of the NT ASO cells are located with the ILF2 ASOs (top panel). Do the Authors know why? Any hypothesis?

In general, the scRNA part requires more explanation, and it has hard to understand. For example, why the Authors claimed: "suggests that persistent exposure to ILF2 ASOs did not induce clonal selection"? The fact that we have two clusters based on two different treatments just suggests that the cells changed their expression according to the different exposure, and doesn't exclude a selection. Have the authors tried to see copy number changes difference between pre and post-treatment? That is probably the best way to support the absence of selection.

Figure 3D: more details about this cohort should be added. It is unclear which cohort we are looking at. How many high-risk and 1q? which treatment they received before SCT? Also, not all curves are significant, and all p-values should be considered (at least between the main groups). Alternatively, Authors can test these features as linear (e.g. cox model)

More details from the patients included in Supplementary Figure S4F should be reported.

The Authors started their story from 1q and then in the discussion they wrote: "Although DNA2 expression was significantly correlated with the 1q21 amplification in MM PCs, DNA2 activity inhibition significantly depleted both 1q21 and non-1q21 amplified PCs from patients that were

refractory to PI-based therapy". Based on these sentences and narrative, it is unclear to me if this potential therapeutic approach is linked to 1q gain/amp or is just a nice and effective way to block high proliferative and aggressive disease. This is essential to clarify and the narrative should be adjusted accordingly.

Minor:

Figures need titles and more details, often it is unclear what we are looking at.

p-values should be added where a test and the word significant is used in the text.

Suppl figures are too crowded and could be divided. Nat comm allows a large number of suppl. figures.

A sentence specifying that myeloma cell lines are not representative of the myeloma seen in patients should be added.

Reviewer #2 (Remarks to the Author):

Thongon et al. present a comprehensive and detailed study demonstrating the utility of ILF2 inhibition in sensitizing myeloma cells to DNA damaging agents. They test ILF2-targeted antisense oligonucleotides (ASO) as a strategy to selectively inhibit ILF2 gene products and probe the molecular mechanisms involved in ILF2 ASO response and resistance. The authors provide mechanistic data from in vitro and in vivo studies along with complementary RNA-seq, metabolomics, and large-scale genomics studies to uncover DNA2 as a mediator of ILF2 ASO resistance. Overall, this is an impressive pre-clinical research effort that has the potential to uncover new classes of clinically-relevant therapies for refractory MM patients, with implications for other cancers that feature DNA2 or ILF2 activation.

Major points:

1. ILF2 is likely activated in MM primarily through amplification of chr1q, as the authors discuss in detail. Would ILF2-targeted therapies only be potentially useful in 1q-amplified patients? Also, the authors may want to discuss the potential clinical relevance of other genes on this locus that may also contribute to disease progression and drug resistance.
2. In the context of the expected 1q21-associated expression of ILF2 expression, do the authors have any data on the amplification status of 1q in the various cell lines tested, and whether this and/or the baseline expression levels of ILF2 correlate with their response to ILF2 ASOs or propensity to develop resistance to the ASOs?
3. The authors introduce a novel potential therapeutic strategy for MM – ILF2 ASOs, which could potentially be used alone or in combination with DNA damaging agents and/or DNA2 inhibitors. It would be helpful if the authors could comment in the Discussion section on what might be the most effective approach to the next steps of development for these agents that would facilitate clinical advancement.

Figures:

Figure 1/S1: For each panel, please clarify which ASO was used. The statistical tests/significance for each panel should also be reported. If no statistically different values were detected for some of the measurements (e.g. panel A), then it would be helpful for this to be stated.

While the panel of ASOs tested in Fig. 1A show no consistent effects on biochemical or histopathological parameters, it appears that ASO 09 may affect ALT levels and liver weights (Fig S1). The authors should comment on this potential effect, since 2 other ASOs tested in Fig. 1 showed similar effects.

It would also be helpful to show at least a representative bioluminescence image of untreated/ASO-treated xenograft mice to show the dissemination of cells in vivo. Were there differences in tumor cell burden in BM of treated mice?

Figure 2/S2, 3/S3: Additional labeling of the different graphs would help ease interpretation of the results (e.g., including cell line or patient group evaluated in each graph, especially when individual panels have multiple subpanels with similarly-labeled graphs showing data from different cell lines/populations).

General/minor formatting comments:

Overall the figures are well laid out and easy to read. The supplemental figures could potentially be divided into separate figures to enhance viewing and interpretation of results. Also, including labels on individual sub-panels to identify different cell lines could further improve the readability of the data.

Line 260 – Does “DNA activity inhibition” refer to “DNA2 activity inhibition”?

Reviewer #3 (Remarks to the Author):

Overall, this is very exciting work that brings the field forward and that increases our understanding of DNA damage therapy resistance for MM patients. The methods are innovative and cutting edge, and the group has done an outstanding amount of work. The figures are beautiful, the discussion section is strong. My critiques and suggestions to improve the paper are below. The results are noteworthy and of great importance to the field. It moves the field forward and the work is original.

Supp Fig 1- Legend title should say in vitro

Supp Fig 1B- You don't mention the result in alanine aminotransaminase- this could be worrying- for the other inhibitors. And in Figure 1A- ALT looks to be very high with the ASOs- could be toxic. Other issues also seen except for in the 09 sample- can you explain why those others didn't look as good in terms of their safety profile?

Why did you have to irradiate NSG mice?

In the methods: “JJN3 cells were treated with 1 μ M NT or ILF2 ASOs for 3 weeks”. Can you explain more about that- was media changed every day or 2X/week etc and how frequently were new ASOs added?

Sup Fig 1B and 1C- should also have dose of ASOs in legend- and throughout the paper in all legends, even though they are written in methods.

Supp Fig 1D clarify in legend what measurement tool was used- how were cells counted?

Supp Fig 1E,F should be 2-way ANOVA since you have 2 variables- drug and ASOs. Then use a post-hoc test to compare groups of interest. Please check this correction of statistics throughout the paper.

Supp Fig 1N- Starting from here, this should be a new sup figure- it is now in vivo work. Also say how they were injected in the legend (IV?) Also, how can you randomize at day 0 on tumor burden? This is usually Very low- please show the BLI images and quantification for the groups you used for randomization. Also it is unclear how many groups this is- Mice were injected with NT or ILF2 ASOs (50 mg/kg) alone for 7 days (day 7) prior to receiving NT or ILF2 ASOs (25 mg/kg) in combination with Melph (2.5 mg/kg) every other day for 3 doses (day 12)- Please tell us each group specifically and in Figs O and P, we want to see all groups. Actually- O is IHC and western blot for quantification? This is very unclear- the IHC is hard to see any clear conclusion from and the WB (graphical data) should also show the blots of the gene and actin. Does less ILF4 mean just less tumor or less ILF4 expression in the tumor? This is all hard to correctly determine- it could be just the healthy/non-tumor cells that have less ILF4, not necessarily the tumors. This line is not supported by your data: “Immunohistochemical analysis showed a 50% reduction in ILF2 levels in KMS11 cells from the bone marrow (BM) and the liver of xenografts treated with ILF2 ASOs in combination with melphalan”.

What did the BLI look like at the 1 week mark of that study- before Melphalan was used. Show us the Flux overtime for all animals and not just at 1 timepoint (Fig 1E). Why didn't you show these? Are you trying to say that Fig 1E is just bone marrow- isn't it actually whole mouse? You didn't look at just BM I don't believe.. so change your results about this. Also why are there so few animals in the liver tumor burden analysis?

Also Fig 1E legend- you say Statistically significant differences were detected using a paired 2-tailed Student t-test ($P = 0.08$). AND Statistically significant differences were detected using an unpaired 2-tailed Student t-test ($**P < 0.01$)"- which is correct?

Fig 2: doses of ASOs should also be written in legends

Scale bars needed in Fig 1 O

Supp Fig 2C- this line is very unclear: "Pathway enrichment analysis of genes that were significantly downregulated in JJN3 but not in KMS11 cells treated with ILF2 ASOs for 3 weeks as compared with those treated for 1 week." Please tell us more about results with KMS11 and not just the JJN3 cells.

Supp Fig 2E- why are there 2 populations of cells in the JJN3 cells? Some inherent grouping it looks like- what is that due to? Can you dig in there and see what the driving genes are?

Supp Fig 2G coloring seems wrong- Is this just raw data? You need to use normalized data.- which makes me question the heatmaps above also. Also, ideally you should use a Zscore for you heat maps as this legend as you have it is not very meaningful.

Your Text in Supp Fig 2I and down is red like a revised manuscript- please double check your text is black on your next version.

Supp Fig 2 – most of these have wrong stats- need 2 way anova.

Supp Fig 2L: Why did ILF2 ASOs alone have the worst outcomes. This in vivo experiment is poorly explained- I can't tell how many groups there are- did the NT before injection mice all stay as the NT after injection groups? More controls should have been used- we can't tell if it's the pre-exposure to ILF2 ASOs or exposure during the study that caused these outcomes. I think your diagram is wrong in Supp Fig 2K and that all the cells were pre-exposed for 3 weeks to ILF2 ASOs, but that's not what this shows.. if not, then they aren't all resistant cells you injected..

Supp Fig 3G legend- it needs to be more clear if quartile 1 is high or low DNA2 expression.. clarify please. Legend should also say where data are from so we don't have to search the methods.

Supp Fig 3I- One way ANOVA is incorrect to use.

Legends are not clear overall. For example, you say things like this often: "Data are expressed as the mean \pm S.D. from one representative experiment"- it is not a "representative" experiment if you only did it once. Did you just do it once? Please make that explicit in every legend.

In Supp Fig 1D it looks like the ILF2 ASOs work on their own in 2 different MM cell lines (they work a little better in KMS11 than JJN3 it looks like). Calculation of EC50 value would be good here in both cell lines to really compare the sensitivity of each. Moreover, this data seems to support that targeting ILF2 with ASOs works on its own, some why doesn't it work well in vivo or why does it have to be combined with targeting DNA2..

In results section- how were 300 ASOs designed or made?

Please state if JJN3 and the KMS line are 1q21 amplification lines or not sooner. Als explain why you picked these 2 cell lines in your results. All your data in Fig S1G-S1M suggest that even NON 1q21 amplified cells are sensitive to ASOs against ILF4 in some cases, and that combination therapies (Bort/melphalan + ILF4 ASOs) work in those cells, so please think carefully about the mechanism of action here and if your conclusions are correct. Why do you see some effects of the ILF2 ASOs even in non 1q21 amplified cells? Do they just happen to have more ILF2 in those MM cell lines?

In Result- this line is out of the blue- you never told us that you made (or how you made or defined) sensitive vs resistant cells. "Consistent with this observation, ILF2 ASO- resistant JJN3 cells were significantly more sensitive to the OXPHOS inhibitor IACS-01075912 than the ILF2 ASO-sensitive cells were (Supplementary Fig. S2H)."

It is often unclear if you are showing data to show the mechanism of how ILF2 ASOs kill cells, or how cells respond to this and become resistant to the effects of the ILF2 ASOs. These need to be better broken up and explained.

Scale bars on Figure 4C must be wrong. It looks like we are zooming in and the top says 200um, then 800um in the middle, but then again a 200um scale of the same size as the top. Also, each box needs a scale bar, not just boxes on the right. Dotted outlines of where you zoomed it would be helpful for us to follow this progression in your imaging. Other differences are seen in your samples- like where is the nucleolus in the NSC treated cells? To be super clear you should also just label top middle and bottom as the magnification they are (7500X, 20,000X, and 50,000X).

This line: "exposed to NSC had upregulated expression of genes involved in respiratory electron transport and ATP synthesis, as an attempt to compensate for the decline in mitochondrial activity and maintain their survival.." – then what did the other cells upregulate to survive?

Minor: You need a comma here: Among patients with the 1q21 amplification who have relapsed the median overall survival duration is a dismal 9 months – in this line (line 78)

- Michaela Reagan

Reviewer #4 (Remarks to the Author):

Building upon the previous novel finding that ILF2 is the driver oncogene in the MM with 1q21 amplification, the authors now developed ILF2-targeting ASO reagents as a potential therapy, and reported involvement of metabolic rewiring and mitochondrial function downstream of ILF2 impairment. Despite utilizing multiple techniques with a large body of data ranging from ASO engineering, bulk and single-cell RNAseq, CRISPR screen and metabolism, following major concerns dampen enthusiasm.

(1) While the authors demonstrated that the ILF2 ASOs developed are effective in the in vitro MM cells confirming the previous finding by RNAi, the in vivo ASO efficacy is not promising. For instance, the ILF2 reduction is barely 50% (Fig. S10); the difference between body tumor burden is not striking (Fig. 1E left); bone marrow tumor burden was not assessed of relevance to MM; liver tumor burden (Fig. 1E right) for NT vs ILF2 ASO treatment without Melph needs to be presented to demonstrate the sensitization effect, rather than ILF2 inhibition -- a sensitization effect shown in the previous study.

(2) The conclusion on the metabolic rewiring with OXPHOS upregulation upon ILF2 ASO requires further validations. For instance, the key metabolism-related experiments need to be performed with at least two independent ILF2 targeting ASOs, or rescuing experiments with ASO-resistant ILF2 construct to confirm due to on-target effect. The conclusion on increased mitochondrial OXPHOS upon ILF2 ASOs is not well supported by the data presented, as ILF2 ASO cells did not exhibit increased basal respiration compared to NT ASO (Fig. 4A).

(3) The CRISPR screen with the screen hit DNA2 and in vivo xenograft result is potentially interesting. The finding needs be strengthened by additional data using genetic perturbation, because all data are based on one inhibitor NSC.

RESPONSE TO REVIEWERS' COMMENTS

Reviewer #1 (Comments to the Authors)

Thongon et al. presented an interesting study showing how myeloma cells' metabolic rewiring is an essential mechanism through which the tumor cell can survive DNA damage. Targeting this mechanism suppresses MM cells' ability to survive DNA damage (chemo-induced) and represents a promising new therapeutic target.

Overall, the paper is well-written and presents several experiments and data in support of the Authors' narrative and hypothesis. Nevertheless, there are aspects of the paper that require careful revision and the Authors should provide more data to support some of their statements.

We thank the Reviewer for the very positive comments regarding the novelty and quality of our study, as well as for the detailed and constructive suggestions that were provided. We hope that our explanations and the additional experiments we included in the revised manuscript improved it and address the Reviewer's well taken concerns.

Major comments:

Comment 1. The terminology needs to be revised. Amplification 1q is used to define 2 or more extra copies of 1q (Walker et al. Leukemia 2019). These patients are also called double-hit. Authors should carefully distinguish between gain 1q (one extra copy) and amp 1q (>1 extra copy) and investigate if their gene signatures and data are affected by a gene dosage effect (e.g. more 1q copies more effect).

Authors' response: We apologize for missing this reference. We have modified the introduction and the text accordingly. We also updated Suppl Figure 3F and changed the labelling accordingly.

The effect of *ILF2* depletion based on copy number gain/amplification was already addressed in our previous article Marchesini et al., *Cancer Cell* 2017;32(1):88. *ILF2* was one of the 78 genes that showed 1q21 copy number driven expression. The sensitivity of MM cells to *ILF2* depletion depends on 1q21 copy number. On the mechanistic level, we previously demonstrated that *ILF2* mediates drug resistance to genotoxic agents in a dose-dependent manner in part by modulating YB-1's nuclear localization and interaction with the splicing factor U2AF65 to promote mRNA processing and stabilization of DNA repair genes in response to DNA damage.

Comment 2. The AST and Liver weight seem to be increased in Figure 1A. Was this significantly higher compared to the controls. Does this data create any conflict with the statement: "We did not observe either consistent histopathological or biochemical ASO-induced alterations, which suggests that *Ilf2* depletion does not induce on-target toxicity (Fig. 1A)."

Authors' response: The ASOs accumulate in the liver, and for this reason, they may induce liver toxicity; these findings are strongly supported by long-standing evidence, for example, Swayze et al., *Nucleic Acids Research* 2007;35(2):687. We would like to please clarify that our conclusion was that *ILF2* depletion does not induce on-target toxicity (and not that the ASOs have no toxicities). Indeed, the ALT (not AST) levels did not increase in mice treated with the antisense 1072209, although *ilf2* expression significantly decreased, which suggests that the ASO backbone, but not the depletion of *ILF2*, may induce an increase in ALT levels.

Comment 3. In general, it would be good to add p-values to all boxplots or generate supplementary tables where the p-value of each comparison is reported.

Authors' response: We have added statistical analysis for each comparison, according to the Reviewer's suggestion.

Comment 4. JJN3 is a MAF translocated cell line. It also has 1q gain. Do the Authors think that the sensitivity to ILF2 ASO is due to 1q, or also to MAF? Or in general to high proliferation and aggressive biology?

Authors' response: This is a well taken question that we addressed in our previous article Marchesini et al., *Cancer Cell* 2017;32(1):88. According to our findings in our previous publication, the effect of *ILF2* depletion is independent of *MAF* translocation and depends on the gain/amplification of 1q21. In this manuscript, we included an extensive characterization of the effect of *ILF2* ASOs in 5 different cell lines and did not aim to discuss targeting *ILF2*; however, we used *ILF2* ASOs as a tool to target *ILF2*, because it is one of the major regulators of response to DNA damage in myeloma. As stated in the introduction, the final goal of our experiments was to discover novel mechanisms of MM cells' resistance to DNA damaging agents in the context of 1q21 gain/amp, and not to study the effects of *ILF2* depletion, which was previously addressed in Marchesini et al., *Cancer Cell* 2017.

Comment 5. It is unclear in the narrative why KMS11 was selected. This cell line has 1q, but were there any other reasons?

Authors' response: ASOs were delivered by free uptake. KMS11 and JJN3 cells were the myeloma cell lines with the best efficiency of free uptake.

Comment 6. Figure 1E: was the difference in liver size significant?

Authors' response: We could not see any significant difference in liver size.

Comment 7. Figure 2C: some of the NT ASO cells are located with the ILF2 ASOs (top panel). Do the Authors know why? Any hypothesis?

Authors' response: The cells mentioned by the Reviewer belong to a specific cluster (cluster 10; please see

Figure on the left), which suggests that the cells in cluster 10 have a similar transcription profile independently of the treatment. Although the quality of these cells was good,

cells in this cluster expressed a signature of apoptosis. It is possible that these cells were undergoing apoptosis independently of the treatment; indeed, they only constituted 1.45% of the total NT ASO-treated cells. Cluster 10 was previously excluded from the differential analysis shown in Figure 2D.

Comment 8. In general, the scRNA part requires more explanation, and it has hard to understand. For example, why the Authors claimed: "suggests that persistent exposure to ILF2 ASOs did not induce clonal selection"? The fact that we have two clusters based on two different treatments just suggests

that the cells changed their expression according to the different exposure and doesn't exclude a selection. Have the authors tried to see copy number changes difference between pre and post-treatment? That is probably the best way to support the absence of selection.

Authors' response: We agree with the Reviewer and performed the copy number analysis that the Reviewer requested (we used the CytoSNP-12 array from Illumina). We did not observe any difference between NT ASO-treated and ILF2 ASO-treated cells after 3 weeks of treatment (n=3 biological replicates per group). We included the results in the new Supplemental Figure 2F.

Comment 9. Figure 3D: more details about this cohort should be added. It is unclear which cohort we are looking at. How many high-risk and 1q? which treatment they received before SCT? Also, not all curves are significant, and all p-values should be considered (at least between the main groups).

Alternatively, authors can test these features as linear (e.g. cox model).

Authors' response: We analyzed the cohort of MM patients who were enrolled in the Arkansas Total Therapy 2 (TT2) trial and received high-dose chemotherapy followed by stem cell transplantation (n=351; *Barlogie et al., Blood 2005*). TT2 was described in detail in the article that we referenced in the Methods section.

Samples with all the relative information are posted at the GEO site

(<https://ftp.ncbi.nlm.nih.gov/geo/series/GSE2nnn/GSE2658/matrix/>).

We provided the Cox proportional hazards model below, according to the Reviewer's suggestion.

Cox proportional hazards model:

Covariate	HR (95% CI, p-value)
DNA2	1.71 (1.25-2.35, p=0.001)

The 1q21 data were available for 248 samples (no amp/gain: n=134; 1q21 gain: n=70; 1q21 amp n=44. DNA2 expression in no amp/gain vs amp/gain: p<0.001 Wilcoxon rank sum test).

Comment 10. More details from the patients included in Supplementary Figure S4F should be reported.

Authors' response: The experiment shown in Figure S4F was performed using plasma cells isolated from 2 healthy donors (HDs) and not BM specimens from patients. The characteristics of these 2 HDs are listed below.

Sample ID	Sex	Age	Race	Blood type	Sample status	Medication use	Donor viral status (HIV, HBV, HCV)	Tobacco use
HD84	Male	54	White	A ⁺	Healthy donor	No	Negative	No
HD77	Male	42	Other race	O ⁻	Healthy donor	No	Negative	No

Comment 11. The Authors started their story from 1q and then in the discussion they wrote: "Although DNA2 expression was significantly correlated with the 1q21 amplification in MM PCs, DNA2 activity inhibition significantly depleted both 1q21 and non-1q21 amplified PCs from patients that were refractory to PI-based therapy". Based on these sentences and narrative, it is unclear to me if this potential therapeutic approach is linked to 1q gain/amp or is just a nice and effective way to block high

proliferative and aggressive disease. This is essential to clarify and the narrative should be adjusted accordingly.

Authors' response: We agree with the Reviewer. The effect of DNA2 activity inhibition on myeloma primary samples after PI-based therapy failure seems to be independent of the 1q21 status. We have modified the text accordingly (please see page 15 of the revised manuscript).

Minor Comments:

Comment 1: Figures need titles and more details, often it is unclear what we are looking at.

Authors' response: We have included titles to clarify which cell line or gene we were referring to.

Comment 2: p-values should be added where a test and the word significant is used in the text.

Authors' response: We have included the p-values in the legends.

Comment 3: Suppl figures are too crowded and could be divided. Nat comm allows a large number of suppl. figures.

Authors' response: We agree with the Reviewer's suggestions about the figures. We will divide the Supplemental figures into separate figures after the revision process according to the guidelines of *Nature Communications* if the manuscript is accepted for publication. We are familiar with the journal guidelines regarding figures from our previous publication (*Thongon N. et al., Nature Communications 2021;12:6850*).

Comment 4: A sentence specifying that myeloma cell lines are not representative of the myeloma seen in patients should be added.

Authors' response: We will include a short paragraph at the end of the discussion in the "Limitations of the study" section if the Reviewer considers this necessary to accept our manuscript. However, we would like to note that a statement about the myeloma cell lines was not included in other publications in which myeloma cell lines were studied.

Reviewer #2 (Comments to the Authors)

Thongon et al. present a comprehensive and detailed study demonstrating the utility of ILF2 inhibition in sensitizing myeloma cells to DNA damaging agents. They test ILF2-targeted antisense oligonucleotides (ASO) as a strategy to selectively inhibit ILF2 gene products and probe the molecular mechanisms involved in ILF2 ASO response and resistance. The authors provide mechanistic data from *in vitro* and *in vivo* studies along with complementary RNA-seq, metabolomics, and large-scale genomics studies to uncover DNA2 as a mediator of ILF2 ASO resistance. Overall, this is an impressive pre-clinical research effort that has the potential to uncover new classes of clinically-relevant therapies for refractory MM patients, with implications for other cancers that feature DNA2 or ILF2 activation.

We greatly appreciate and would like to sincerely thank the Reviewer for the highly commendatory comments they provided regarding the novelty, high quality, and impact of our preclinical research studies in the field of refractory MM. We would also like to sincerely thank the Reviewer for providing meaningful questions and constructive suggestions, which have definitively improved our manuscript. We addressed the comments below and hope that the Reviewer finds our responses and the revisions that we made to the manuscript satisfactory.

Major points:

Comment 1. ILF2 is likely activated in MM primarily through amplification of chr1q, as the authors discuss in detail. Would ILF2-targeted therapies only be potentially useful in 1q-amplified patients? Also, the authors may want to discuss the potential clinical relevance of other genes on this locus that may also contribute to disease progression and drug resistance.

Authors' response: We have discussed other genes that account for 1q21 amplification-induced bad prognosis in our previous article, *Marchesini et al., Cancer Cell 2017;32(1):88*. In this study, we conducted a genetic screening to find novel target genes in 1q21, whose expression was copy-number driven. *MCL1* was definitively one of the genes. However, since this finding was already known in the field, we excluded *MCL1* from further validations. In the present manuscript, we focused on targeting *ILF2* using ASOs as a tool to induce DNA damage activation. We did not focus on genes in 1q21 that accounts for the 1q21 high risk phenotype because we discussed this in our previous study: "Increased *ILF2* expression was strongly correlated with poorer survival in MM patients treated with high-dose melphalan followed by tandem autologous transplantation." However, we recognize that additional 1q21-amplified and overexpressed genes, such as *MCL1*, or other genes that could be missed by our *in vitro* screening, may contribute to MM prognosis.

Comment 2. In the context of the expected 1q21-associated expression of ILF2 expression, do the authors have any data on the amplification status of 1q in the various cell lines tested, and whether this and/or the baseline expression levels of ILF2 correlate with their response to ILF2 ASOs or propensity to develop resistance to the ASOs?

Authors' response: We included the gain/amplification status of the cell lines used in the manuscript in the Table below (based on the reviewer's comment we run the 1p32/1q21 FISH analysis to confirm previous data by *Hanamura et al., Blood 2006*). It is impossible to conclude whether there is a correlation with the expression

A	
myeloma cell lines	1q21 copies (% clonal size)
H929	4 (18,5%), 5 (56,5%), 6 (18%), 8 (7%)
JJN3	4 (80%), 8 (20%)
KMS11	8 (100%)
MM1R	3 (85%), 4 (12%), 6 (3%)
RPMI-8266	3 (16%), 4 (71%), 5 (4%), 7 (7%)

B		
myeloma cell lines	analyzed nuclei	hybridization pattern
H929	200	37n:2G4O, 113n:2G5O, 36n:2G6O, 14n:4G8O
JJN3	200	160n:3G4O, 40n:6G8O
KMS11	100	1G8O
MM1R	100	85n:2G3O, 12n:2G4O, 3n:4G6O
RPMI-8266	100	71n:2G4O, 16n:2G3O, 7n:2G7O, 4n:2G5O

Frequencies of 1q21 copies (**A**) and hybridization patterns (**B**) in H929, JJN3, KMS11, MM1R, and RPMI-8266 cells evaluated by FISH analysis using the XL 1p32/1q21Amplification/Deletion Probe (Metasystem). N, nuclei; G, green signal detecting the 1q32 locus; O, orange signal detecting the 1q21 locus (CKS1B).

level of *ILF2* and the propensity to develop resistance to the ASOs. MM1R never developed resistance to ASOs although MM1R cells harbor 3 1q21 copies in 85% of the cells.

Comment 3. The authors introduce a novel potential therapeutic strategy for MM – ILF2 ASOs, which could potentially be used alone or in combination with DNA damaging agents and/or DNA2 inhibitors. It would be helpful if the authors could

comment in the Discussion section on what might be the most effective approach to the next steps of development for these agents that would facilitate clinical advancement.

Authors’ response: We are in the process of developing small molecules to target DNA2’s (mitochondrial) nuclease activity. We think that targeting DNA2 activity (without any combination) can also overcome metabolic rewiring in other cancers if they depend on oxidative phosphorylation to maintain survival after DNA damaging therapies. We briefly discussed this point in the “Conclusions” section of the manuscript.

Figures:

Comment 4. Figure 1/S1: For each panel, please clarify which ASO was used. The statistical tests/significance for each panel should also be reported. If no statistically different values were detected for some of the measurements (e.g. panel A), then it would be helpful for this to be stated.

Authors’ response: The Reviewer’s comments on the figures are well taken. We have included the following sentence in the text: “The ILF2 ASO 1146809 (09), which elicited the best dose response and had an acceptable tolerability profile in mice was selected for all functional validation studies in MM cells”. We have added the statistical analyses in all the plots according to the Reviewer’s suggestion.

Comment 5. While the panel of ASOs tested in Fig. 1A show no consistent effects on biochemical or histopathological parameters, it appears that ASO 09 may affect ALT levels and liver weights (Fig S1). The authors should comment on this potential effect, since 2 other ASOs tested in Fig. 1 showed similar effects.

Authors’ response: The ASOs accumulate in the liver, and for this reason, they may induce liver toxicity; these findings are strongly supported by long-standing evidence, for example, Swayze et al., *Nucleic Acids Research* 2007, 35(2):687. Our conclusion was that *ilf2* depletion does not induce on-target toxicity (and not that the ASOs have no toxicities). Indeed, the ALT levels did not increase in mice treated with the antisense 1072209, although *ilf2* expression significantly decreased. This means that the backbone of the ASOs and not the downregulation of *ilf2* induced on-target toxicity.

Comment 6. It would also be helpful to show at least a representative bioluminescence image of untreated/ASO-treated xenograft mice to show the dissemination of cells in vivo. Were there differences in tumor cell burden in BM of treated mice?

Authors' response: We did not observe any difference in terms of bioluminescence between NT and ILF2 ASO-treated mice. We have included these data in the new Figure 1E. Antisense uptake in BM MM cells was low, as demonstrated by the partial reduction of ILF2 in BM MM cells. The partial decrease of *ILF2* expression did not induce caspase 3 activation (updated Supplementary Figure 1O), and it is only effective in combination with melphalan.

Comment 7. Figure 2/S2, 3/S3: Additional labeling of the different graphs would help ease interpretation of the results (e.g., including cell line or patient group evaluated in each graph, especially when individual panels have multiple subpanels with similarly-labeled graphs showing data from different cell lines/populations).

Authors' response: We followed the Reviewer's suggestions and updated the figures accordingly.

General/minor formatting comments:

Overall, the figures are well laid out and easy to read. The supplemental figures could potentially be divided into separate figures to enhance viewing and interpretation of results. Also, including labels on individual sub-panels to identify different cell lines could further improve the readability of the data.

Authors' response: We would like to thank the Reviewer for the positive comments regarding the legibility of the figures and the constructive suggestions to enhance the clarity of the figures. We will divide the Supplemental figures into separate figures after the revision process, according to the guidelines of *Nature Communications* if the manuscript is accepted for publication. We are familiar with the journal guidelines regarding figures from our previous publication (*Thongon N. et al., Nature Communications 2021;12:6850*).

Line 260 – Does “DNA activity inhibition” refer to “DNA2 activity inhibition”?

Authors' response: We agree with the Reviewer and apologize for this mistake. We have corrected it in the manuscript.

Reviewer #3 (Comments to the Authors)

Overall, this is very exciting work that brings the field forward and that increases our understanding of DNA damage therapy resistance for MM patients. The methods are innovative and cutting edge, and the group has done an outstanding amount of work. The figures are beautiful, the discussion section is strong. My critiques and suggestions to improve the paper are below. The results are noteworthy and of great importance to the field. It moves the field forward and the work is original.

We truly appreciate and thank Dr. Reagan for her praising comments regarding the impact and novelty of our manuscript and the high quality of our research. We would also like to sincerely thank Dr. Reagan for her detailed questions and suggestions, which have definitively improved our manuscript. We addressed the comments below and hope that Dr. Reagan finds our responses and the revisions we made to the manuscript satisfactory.

Comment 1: Supp Fig 1- Legend title should say in vitro.

Authors' response: We will divide Supplementary Figure 1 and modify the title of Figure 1 if our article is accepted for publication.

Comment 2: Supp Fig 1B- You don't mention the result in alanine aminotransaminase- this could be worrying- for the other inhibitors. And in Figure 1A- ALT looks to be very high with the ASOs- could be toxic. Other issues also seen except for in the 09 sample- can you explain why those others didn't look as good in terms of their safety profile?

Authors' response: The ASOs accumulate in the liver, and for this reason, they may induce liver toxicity; these findings are strongly supported by long-standing evidence, for example, Swayze et al., *Nucleic Acids Research* 2007;35(2):687. Our conclusion was that *ILF2* depletion does not induce on-target toxicity (and not that the ASOs have no toxicities). Indeed, the ALT levels did not increase in mice treated with the antisense 1072209 (Figure 1a), although *Ilf2* expression significantly decreased. In other words, an increase in ALT levels is normal in the setting of antisense delivery.

Comment 3: Why did you have to irradiate NSG mice?

Authors' response: We sublethally irradiated NSG mice to enhance the human cell engraftment. When we transplanted KMS11 cells without irradiating the recipient mice, we could not observe any engraftment after a few months.

Comment 4: In the methods: "JJN3 cells were treated with 1 μM NT or ILF2 ASOs for 3 weeks". Can you explain more about that- was media changed every day or 2X/week etc and how frequently were new ASOs added?

Authors' response: NT or ILF2 ASOs were added together with fresh media every 2 days. We added this sentence in the "Methods" section.

Comment 5: Sup Fig 1B and 1C- should also have dose of ASOs in legend- and throughout the paper in all legends, even though they are written in methods.

Authors' response: We included this information in the legends.

Comment 6: Supp Fig 1D clarify in legend what measurement tool was used- how were cells counted?

Authors' response: Cells were counted under the microscope using Trypan blue staining. We added this information in the legend.

Comment 7: Supp Fig 1E, F should be 2-way ANOVA since you have 2 variables- drug and ASOs. Then use a post-hoc test to compare groups of interest. Please check this correction of statistics throughout the paper.

Authors' response: We modified the statistics in Suppl. Figures 1E and 1F based on the reviewer's great suggestion.

The Tukey's honest significant test was used for post hoc pairwise comparisons between different treatments.

Comment 8: Supp Fig 1N- Starting from here, this should be a new sup figure- it is now in vivo work.

Authors' response: We thank Dr. Reagan for the suggestion regarding the figures. We will divide the Supplemental figures into separate figures after the revision process, according to the guidelines of *Nature Communications* if the manuscript is accepted for publication. We are familiar with the journal guidelines regarding figures from our previous publication (*Thongon N. et al. Nature Communications 2021;12:6850*).

Comment 9: Also say how they were injected in the legend (IV?)

Authors' response: We included the information in the legend.

Comment 10: Also, how can you randomize at day 0 on tumor burden? This is usually Very low- please show the BLI images and quantification for the groups you used for randomization.

Authors' response: We randomized the mice 10 days after transplantation based on the bioluminescence signal. We added this information in the legend. Representative BLI images of NSG mice transplanted with GFP⁺Luc⁺KMS11 cells after transplantation are shown in the Figure on the left.

Comment 11: Also, it is unclear how many groups this is- Mice were injected with NT or ILF2 ASOs (50 mg/kg) alone for 7 days (day 7) prior to receiving NT or ILF2 ASOs (25 mg/kg) in combination with Melph (2.5 mg/kg) every other day for 3 doses.

Authors' response: We have included the number of mice per group in the legend of Figure S1N (NT ASOs+Veh, n=17; NT ASOs+Melph, n=16; ILF2 ASOs+Veh, n=16; ILF2 ASOs+Melph, n=14. Two independent experiments were performed.

Comment 12: Please tell us each group specifically and in Figs O and P, we want to see all groups. Actually- O is IHC and western blot for quantification? This is very unclear- the IHC is hard to see any clear conclusion from and the WB (graphical data) should also show the blots of the gene and actin.

Authors' response: We have included the IHC analyses relative to the 4 groups.

We would like to note that we did not mention western blot analysis in Supplementary Figure 1O. The quantification was performed using real-time PCR (we included this in the legend) in GFP⁺ sorted MM cells from the BM of the mice. Because melphalan reduced the tumor burden, we could not isolate enough cells to perform the same analysis in melphalan-treated mice.

Comment 13: Does less ILF4 mean just less tumor or less ILF4 expression in the tumor?

Authors' response: We would like to clarify that expression of *ILF2* (not *ILF4*) was quantified in the tumor. Suppl. Figure 1O (right panel) stated that we performed real-time PCR in GFP⁺ cells sorted from the BM of the treated mice.

Comment 14: This is all hard to correctly determine- it could be just the healthy/non-tumor cells that have less *ILF4*, not necessarily the tumors. This line is not supported by your data:

“Immunohistochemical analysis showed a 50% reduction in *ILF2* levels in KMS11 cells from the bone marrow (BM) and the liver of xenografts treated with *ILF2* ASOs in combination with melphalan”.

Authors' response: As we clarified in the previous response, expression of *ILF2* and not *ILF4* was quantified in the tumor. We respectfully disagree with Dr. Reagan on this point given that ASOs cannot target mouse cells and the antibody against *ILF2* cannot stain mouse cells.

Moreover, we would like to note that the biopsies were evaluated by 2 pathologists at the MD Anderson Cancer Center and that it is possible to recognize MM cells based on their shape and size, even without staining.

Comment 15a: What did the BLI look like at the 1 week mark of that study- before Melphalan was used. Show us the Flux overtime for all animals and not just at 1 timepoint (Fig 1E). Why didn't you show these? Are you trying to say that Fig 1E is just bone marrow- isn't it actually whole mouse?

Authors' response: We updated Figure 1E. No, the legend of Figure 1E states: “tumoral burden”. We modified the text accordingly.

Comment 15b: You didn't look at just BM I don't believe. so change your results about this. Also why are there so few animals in the liver tumor burden analysis?

Authors' response: We updated the text accordingly. *ILF2* ASOs + melphalan-treated mice had nearly no liver tumoral burden. Thus, we did not analyze more liver biopsies.

Comment 16: Also, Fig 1E legend- you say Statistically significant differences were detected using a paired 2-tailed Student t-test ($P = 0.08$). AND Statistically significant differences were detected using an unpaired 2-tailed Student t-test ($**P < 0.01$)- which is correct?

Authors' response: We changed Figure 1E and updated the statistical analyses accordingly.

Comment 17: Scale bars needed in Fig 1O.

Authors' response: We have included the scale bars in Figure 1O.

Comment 18: Supp Fig 2C- this line is very unclear: “Pathway enrichment analysis of genes that were significantly downregulated in JJN3 but not in KMS11 cells treated with *ILF2* ASOs for 3 weeks as compared with those treated for 1 week.” Please tell us more about results with KMS11 and not just the JJN3 cells.

Authors' response: We would like to clarify that the results were not for the JJN3 cell line; the results were relative to genes that were downregulated in the JJN3 cell line only (and not in KMS11). In other words, we excluded all genes downregulated in both cell lines. We can provide the raw data if Dr. Reagan would like to review the data in order to accept the manuscript for publication.

Comment 19: Supp Fig 2E- why are there 2 populations of cells in the JJN3 cells? Some inherent grouping it looks like- what is that due to? Can you dig in there and see what the driving genes are?

Authors' response: We performed differential analyses between the 2 expression-driven groups in NT ASOs-treated cells (Figure 2C). The results shown in the Figure below suggest that the main cluster on the right is driven by cell cycle genes (*BIRC5*, *CDK1*, *CDKN2A*, *CDKN3*, *CENPE*, *CKS1B*, *DCK*, *H2AX*, *H2AZ1*, *HMGB2*, *HMGA1*, *STMN1*, *LBR*, *MCM3*, *MKI67*, *MYBL2*, *PCNA*, *RAD21*, *RRM2*, *TMPO*, *TOP2A*, *USP1*, *DEK*, *SMC3*,

CCNB2, *AURKB*, *NCAPD2*, *SMC4*, *PRDX4*, *RAD51AP1*, *CTCF*, *MTHFD2*, *KIF2C*, *PSIP1*, *UBE2S*, *ATAD2*, *ANP32E*, *TUBB*, *BUB1*, *CCNA2*, *CENPF*, *E2F1*, *E2F2*, *HMG2*, *SQLE*, *SMC2*, *TPX2*, *NUSAP1*, *SLC38A1*, and *H2AZ2*).

Comment 20: Supp Fig 2G coloring seems wrong- Is this just raw data? You need to use normalized data.- which makes me question the heatmaps above also. Also, ideally you should use a Zscore for you heat maps as this legend as you have it is not very meaningful.

Authors' response: We carefully checked the heatmap in Figure S2G (now S2H), and the colors are correct. All the data were normalized to NT ASOs as reported at the side of the heatmap "Relative to NT ASOs".

Comment 21: Your Text in Supp Fig 2I and down is red like a revised manuscript- please double check your text is black on your next version.

Authors' response: We would like to note that the Editor requested the revisions highlighted in red. Before sending out the manuscript to the Reviewers, the Editor asked us to do the experiments that were highlighted in red in the version that the Reviewers received. After accepting our revision, the Editor sent the revised manuscript directly to the Reviewers, so we did not have the opportunity to change the color of the text.

Comment 22: Supp Fig 2 – most of these have wrong stats- need 2 way anova.

Authors' response: We corrected the statistical analyses accordingly. Changes in the statistical analyses are highlighted in red in the updated manuscript.

Comment 23: Supp Fig 2L: Why did ILF2 ASOs alone have the worst outcomes. This in vivo experiment is poorly explained- I can't tell how many groups there are- did the NT before injection mice all stay as the NT after injection groups? More controls should have been used- we can't tell if it's the pre-exposure to ILF2 ASOs or exposure during the study that caused these outcomes. I think your diagram is wrong in Supp Fig 2K and that all the cells were pre-exposed for 3 weeks to ILF2 ASOs, but that's not what this shows. if not, then they aren't all resistant cells you injected.

Authors' response: The experiment was done using ILF2 ASO-resistant cells as stated in the legend of the previous Suppl Fig 2K (current Suppl. Figure 2L). At the time of injection, the JJN3 cells had already been treated with ASOs for more than 3 weeks. Accordingly, ILF2-ASO-treated mice did not have any difference in survival compared with NT ASO-treated mice. We would like to respectfully note that the graphical abstract in the new Suppl. Figure 2L is correct, and the legend is clear, in our opinion. The legend of Figure 2K reads as follows: "Schematic of ASO and IACS-010759 treatments in MM xenografts. ILF2 ASO-resistant GFP⁺Luc⁺ JJN3 cells were injected into NSG mice. Five days after transplantation, mice were randomized into 4 groups and treated with NT or ILF2 ASOs alone (25 mg/kg) or in combination with IACS-010759 (IACS; 10 mg/kg) on a 5-days-on, 2-days-off cycle until they were euthanized because they were moribund". There were 4 groups of mice in this experiment, as we reported in the legend of Suppl. Figure 2L: "Survival curves of NSG mice that

received transplants of ILF2 ASO-resistant JJN3 cells after receiving NT or ILF2 ASOs alone (NT or ILF2+Veh) or in combination with IACS-010759 (IACS; 10 mg/kg) (NT ASOs+Veh, n=4; NT ASOs+IACS, n=4; ILF2 ASOs+Veh, n=6; ILF2 ASOs+IACS, n=8)".

Comment 24: Supp Fig 3G legend- it needs to be more clear if quartile 1 is high or low DNA2 expression. Clarify please. Legend should also say where data are from so we don't have to search the methods.

Authors' response: We updated the legend accordingly and apologize for the omission.

Comment 25: Supp Fig 3I- One way ANOVA is incorrect to use.

Legends are not clear overall. For example, you say things like this often: "Data are expressed as the mean \pm S.D. from one representative experiment"- it is not a "representative" experiment if you only did it once. Did you just do it once? Please make that explicit in every legend.

Authors' response: We corrected the statistical analyses accordingly and included the number of experiments that we performed in the legend of every figure.

Comment 26: In Supp Fig 1D it looks like the ILF2 ASOs work on their own in 2 different MM cell lines (they work a little better in KMS11 than JJN3 it looks like). Calculation of EC50 value would be good here in both cell lines to really compare the sensitivity of each. Moreover, this data seems to support that targeting ILF2 with ASOs works on its own, some why doesn't it work well in vivo or why does it have to be combined with targeting DNA2.

Authors' response: We would like to clarify that we validated DNA2 in the context of resistance to ILF2 ASOs. ILF2 ASOs were clearly effective in every 1q21-amplified MM cell line that we tested, as we demonstrated in our previous article about ILF2, Marchesini et al., *Cancer Cell* 2017;32(1):88.

The ASOs were delivered by free uptake. KMS11 cells have a better free uptake than JJN3 as demonstrated by the fact that JJN3 cells need a higher concentration of ILF2 ASOs to reach an ILF2 downregulation that is comparable with that in KMS11 (1 μ M vs 0.5 μ M).

Antisense uptake in BM MM cells was low, as demonstrated by the partial reduction of ILF2 in BM MM cells. The partial decrease of *ILF2* expression did not induce caspase 3 activation (updated Supplementary Figure 1P), and it was only effective in combination with melphalan.

Comment 27: In results section- how were 300 ASOs designed or made?

Authors' response: ILF2 ASOs were designed and developed from IONIS Pharmaceuticals, and the process was included in the text on page 6: "ASOs with constrained ethyl chemistry, which induces improved stability, RNA affinity, and resistance against nuclease-mediated metabolism, resulting in a significantly improved tissue half-life in vivo and a longer duration of action".

Comment 28: Please state if JJN3 and the KMS line are 1q21 amplification lines or not sooner. Als explain why you picked these 2 cell lines in your results.

Authors' response: We included the 1q21 status of the cell lines on page 7.

As we stated in the manuscript, ASOs were delivered by free uptake. KMS11 and JJN3 cells were myeloma cell lines with the best efficiency of free uptake.

Comment 29: All your data in Fig S1G-S1M suggest that even NON 1q21 amplified cells are sensitive to ASOs against ILF4 in some cases, and that combination therapies (Bort/melphalan + ILF4 ASOs) work in those cells, so please think carefully about the mechanism of action here and if your conclusions

are correct. Why do you see some effects of the ILF2 ASOs even in non 1q21 amplified cells? Do they just happen to have more ILF2 in those MM cell lines?

A	myeloma cell lines	1q21 copies (% clonal size)	
	H929	4 (18,5%), 5 (56,5%), 6 (18%), 8 (7%)	
	JJN3	4 (80%), 8 (20%)	
	KMS11	8 (100%)	
	MM1R	3 (85%), 4 (12%), 6 (3%)	
	RPMI-8266	3 (16%), 4 (71%), 5 (4%), 7 (7%)	

B	myeloma cell lines	analyzed nuclei	hybridization pattern
	H929	200	37n:2G4O, 113n:2G5O, 36n:2G6O, 14n:4G8O
	JJN3	200	160n:3G4O, 40n:6G8O
	KMS11	100	1G8O
	MM1R	100	85n:2G3O, 12n:2G4O, 3n:4G6O
	RPMI-8266	100	71n:2G4O, 16n:2G3O, 7n:2G7O, 4n:2G5O

Frequencies of 1q21 copies (A) and hybridization patterns (B) in H929, JJN3, KMS11, MM1R, and RPMI-8266 cells evaluated by FISH analysis using the XL 1p32/1q21Amplification/Deletion Probe (Metasystem). N, nuclei; G, green signal detecting the 1q32 locus; O, orange signal detecting the 1q21 locus (CKS1B).

Authors' response:

All the cell lines that we used in our studies had the 1q21 gain/amplification (*Hanamura et al. Blood, 2006; 106 (11):1553*). We also performed 1p32/1q21 FISH analysis to confirm the previous published data. The results for the cell lines are shown on the left.

Comment 30: “Consistent with this observation, ILF2 ASO– resistant JJN3 cells were significantly more sensitive to the OXPHOS

inhibitor IACS-01075912 than the ILF2 ASO–sensitive cells were (Supplementary Fig. S2H)”. It is often unclear if you are showing data to show the mechanism of how ILF2 ASOs kill cells, or how cells respond to this and become resistant to the effects of the ILF2 ASOs. These need to be better broken up and explained.

Authors' response: We agree with Dr. Reagan and corrected the text in the manuscript. We wrote “ILF2 ASO-resistant cells” instead of ILF2 ASO-treated cells to further clarify the experiments (please see page 10).

Comment 31: Scale bars on Figure 4C must be wrong. It looks like we are zooming in and the top says 200um, then 800um in the middle, but then again a 200um scale of the same size as the top. Also, each box needs a scale bar, not just boxes on the right. Dotted outlines of where you zoomed it would be helpful for us to follow this progression in your imaging.

Authors' response: We agree with Dr. Reagan and thank her for pointing out the correction. We have corrected the scale bars. We previously included the magnification in the legend. We included the dotted outlines in the figure.

Comment 32: Other differences are seen in your samples- like where is the nucleolus in the NSC treated cells?

Authors' response: We thank Dr. Reagan for asking us a very interesting question! After careful revision of the literature, we found that the nucleolus may undergo segmentation under a stress response as that induced by mitochondrial damage (Latonen L., *Frontiers in Cellular Neuroscience*, 2019;13:151).

Comment 33: This line: “exposed to NSC had upregulated expression of genes involved in respiratory electron transport and ATP synthesis, as an attempt to compensate for the decline in mitochondrial activity and maintain their survival.” – then what did the other cells upregulate to survive?

Authors' response: The other cells (NT ASOs and NSC treated cells) are not sensitive to NSC because they do not depend on mitochondria functions to survive, thus they do not need to compensate mitochondrial activity.

Minor comment: You need a comma here: Among patients with the 1q21 amplification who have relapsed the median overall survival duration is a dismal 9 months – in this line (line 78)

Authors' response: We added the comma.

Reviewer #4 (Comments to the Authors):

Building upon the previous novel finding that ILF2 is the driver oncogene in the MM with 1q21 amplification, the authors now developed ILF2-targeting ASO reagents as a potential therapy, and reported involvement of metabolic rewiring and mitochondrial function downstream of ILF2 impairment. Despite utilizing multiple techniques with a large body of data ranging from ASO engineering, bulk and single-cell RNAseq, CRISPR screen and metabolism, following major concerns damper enthusiasm.

We would like to sincerely thank the Reviewer for providing positive comments as well as constructive criticism and suggestions. We have provided responses to the comments and hope that our explanations address the Reviewer's concerns in a satisfactory manner.

Comment 1. While the authors demonstrated that the ILF2 ASOs developed are effective in the in vitro MM cells confirming the previous finding by RNAi, the in vivo ASO efficacy is not promising. For instance, the ILF2 reduction is barely 50% (Fig. S1O); the difference between body tumor burden is not striking (Fig. 1E left); bone marrow tumor burden was not assessed of relevance to MM; liver tumor burden (Fig. 1E right) for NT vs ILF2 ASO treatment without Melph needs to be presented to demonstrate the sensitization effect, rather than ILF2 inhibition -- a sensitization effect shown in the previous study.

Authors' response: We have now included the results relative to the mice treated with NT and ILF2 ASOs in Figure 1E. Antisense uptake in BM MM cells is low, as demonstrated by the partial reduction of ILF2 in BM MM cells. The partial decrease of *ILF2* expression did not induce caspase 3 activation (updated Supplementary Figure 1P), and it is only effective in combination with melphalan. KMS11 cells have 8 copies of 1q21, which suggests that a decrease in ILF2 expression by 50% is not sufficient to induce DNA damage activation. However, we would like to clarify that we used the ASOs as a tool to study the DNA damage response, and we are not proposing to use ASOs for clinical applications.

Comment 2. The conclusion on the metabolic rewiring with OXPHOS upregulation upon ILF2 ASO requires further validations. For instance, the key metabolism-related experiments need to be performed with at least two independent ILF2 targeting ASOs, or rescuing experiments with ASO-resistant ILF2 construct to confirm due to on-target effect.

Authors' response: We would like to bring to the Reviewer's attention that we diligently screened more than **300 ASOs** in collaboration with IONIS Pharmaceuticals, and only one ASO did not induce "out-of-target" effects *in vitro* and *in vivo*; these studies took one year to complete.

Regarding the rescue experiment with an ASO-resistant ILF2 construct, we do not consider this study necessary given that only the JLN3 cell line became resistant to ILF2 ASO treatment out of the 5 cell lines that we tested. In our opinion the reviewers' request is not supported by our results.

The conclusion on increased mitochondrial OXPHOS upon ILF2 ASOs is not well supported by the data presented, as ILF2 ASO cells did not exhibit increased basal respiration compared to NT ASO (Fig. 4A).

Authors' response: We respectfully disagree with the Reviewer regarding this comment. In Fig. 4A and Suppl Fig. 4A, we demonstrated that ILF2 ASOs (in blue) showed a significant increase of OCR compared to NT ASOs (in grey).

Comment 3. The CRISPR screen with the screen hit DNA2 and in vivo xenograft result is potentially

interesting. The finding needs be strengthened by additional data using genetic perturbation, because all data are based on one inhibitor NSC.

Authors' response: We respectfully disagree with the Reviewer because we tested 2 different DNA2 inhibitors and acquired similar results (please see Suppl. Figure 3I). We also do not consider it necessary to perform a genetic validation experiment because inhibitors that target DNA2 activity and not DNA2 expression may have a clinical application, as stated in the conclusion of the manuscript. However, we understand the need to demonstrate that NSC has an "on-target effect" on the function of mitochondria. Thus, we used Rho wild-type and Rho-0 HL-60 cells without mitochondrial DNA and oxidative phosphorylation to demonstrate that NSC has no impact on cell viability in the context of DNA damage activation. As shown in the Figure below, we could not observe any effect of NSC on Rho-0 cell's ROS production and apoptosis. These findings strongly confirmed that NSC (DNA2 activity inhibition) has on-target effects and affects the survival of cells based on the function of mitochondria.

REVIEWER COMMENTS

Reviewer #1 (Remarks to the Author):

The Authors provided comprehensive and appropriate responses to all of my questions and comments.

Here are a few minor suggestions for improvement:

I would add details about cluster 10 enrichment for apoptotic genes in Figure 2C legend. This will provide additional clarity and help readers understand the significance of this cluster.

The TT2 trial is quite old, and while it makes sense to use it because it was chemo-based, it might be useful to consider incorporating more updated/recent data sets such as CoMMpass. Specifically, the Authors could focus their analysis on patients treated with VRV+ASCT to provide a more recent and relevant perspective.

It is important to always acknowledge and mention the limitations of working with cell lines in myeloma. Including a statement about the limitations of cell lines in the discussion or methodology section will help provide a balanced view of the study's findings.

Reviewer #2 (Remarks to the Author):

The authors have performed a comprehensive revision of their manuscript and have addressed my prior comments in detail as well as the vast majority of the other reviewers' comments. Their revised figures and labeling have improved the clarity of the manuscript, and the gain/amplification status analysis of the cell lines strengthens this revised version of their work and supports their conclusions. In several responses, the authors indicate that they will revise the figures as suggested by the reviewers, if the manuscript is accepted for publication. I recommend that the editorial staff verify that this is completed at that stage.

Reviewer #3 (Remarks to the Author):

The authors have addressed all my prior concerns.

Reviewer #4 (Remarks to the Author):

The demonstration of ASOs as a superior tool for studying the DNA damage response compared to the previously employed RNAi strategy was not clearly established.

Due to the high sensitivity of metabolic changes, it is essential to validate the key metabolic/mitochondrial findings, such as OCR (oxygen consumption rate), using two independent methods (two ASOs, RNAi, or pharmacological approaches). Alternatively, the findings should be supported by rescue experiments.

The response provided regarding the interpretation of the OCR data was unsatisfactory. In Figure 4A, the basal respiration (OCR) remains unaffected between ILF2 ASOs and NT ASOs, whereas only the FCCP-mediated maximum OCR shows a significant increase with ILF2 ASOs. If these findings were indeed validated using another ILF2 perturbation condition, the data would suggest an impairment of the spare capacity of OXPHOS rather than a basal mitochondrial function.

RESPONSE TO REVIEWERS' COMMENTS

REVIEWER #1

The Authors provided comprehensive and appropriate responses to all of my questions and comments.

We thank the reviewer for the positive feedback and suggestions.

Here are a few minor suggestions for improvement:

1) I would add details about cluster 10 enrichment for apoptotic genes in Figure 2C legend. This will provide additional clarity and help readers understand the significance of this cluster.

Based on the reviewer's suggestion, we have included the details relative to cluster 10 in the legend of Figure 2C.

2) The TT2 trial is quite old, and while it makes sense to use it because it was chemo-based, it might be useful to consider incorporating more updated/recent data sets such as CoMMpass. Specifically, the Authors could focus their analysis on patients treated with VRV+ASCT to provide a more recent and relevant perspective.

We have included the analysis requested by the reviewer in the new Figure 3D (left panel). We analyzed the correlation between *DNA2* expression by RNA-seq and the overall survival of 241 MM patients treated with VRD followed by ASCT (VRD = velcade/revlimid and dexamethasone-based therapy) included in the CoMMpass database. This analysis confirmed that high *DNA2* expression predicts a worse overall survival after VRD + ASCT (Log-rank p value = 8.758×10^{-05}).

3) It is important to always acknowledge and mention the limitations of working with cell lines in myeloma. Including a statement about the limitations of cell lines in the discussion or methodology section will help provide a balanced view of the study's findings.

We have included the statement that the reviewer suggested in the method section (page 34, in red).

REVIEWER #4

The demonstration of ASOs as a superior tool for studying the DNA damage response compared to the previously employed RNAi strategy was not clearly established.

We respectfully disagree with the reviewer. We never stated that ASOs are a superior tool for studying the DNA damage response compared with the previously employed RNAi strategies. We used ASOs because of their potential therapeutic use in myeloma (IONIS has ongoing clinical trials of ASOs targeting IRF4).

Due to the high sensitivity of metabolic changes, it is essential to validate the key metabolic/mitochondrial findings, such as OCR (oxygen consumption rate), using two independent methods (two ASOs, RNAi, or pharmacological approaches). Alternatively, the findings should be supported by rescue experiments.

Based on the reviewer suggestion, we have validated the main key metabolic/mitochondrial findings, such as ROS production and OCR using one of the two shRNAs targeting ILF2, previously validated in our 2017 Cancer Cell manuscript (*Marchesini et al., Cancer Cell 2017*). In our previous manuscript, we used 2 different shRNAs targeting ILF2 and showed that ILF2 depleted cells significantly upregulated the expression of phospho-H2AX and increased caspase 3 mediated apoptosis. Here, we repeated the same experiment, but this time we cultured the cells for more than 3 weeks until they became resistant to ILF2 depletion (as shown by either no phospho-H2AX upregulation or increased apoptosis in physiological conditions;

Figures 1 A and B). NSC treatment induced significantly increased apoptosis in ILF2 shRNA-resistant cells compared with those treated with NS shRNAs (**Figure 1C, left panel**). ILF2 shRNA-resistant cells, as those resistant to ILF2 ASOs (Figure 4B in the manuscript)

showed increased ROS production under physiological conditions (vehicle treatment) and after NSC treatment (**Figure 1C, right panel**).

Maximal OCR of NS or ILF2-shRNA infected cells (**Figure 2**) were consistent with those of NS or ILF2 ASOs cells (Suppl. Figure 2J and Figure 4A in the manuscript), which suggests that ILF2 depletion induces an impairment of spare capacity, as also stated by the reviewer in the comment below.

The response provided regarding the interpretation of the OCR data was unsatisfactory. In Figure 4A, the basal respiration (OCR) remains unaffected between ILF2 ASOs and NT ASOs, whereas only the FCCP-mediated maximum OCR shows a significant increase with ILF2 ASOs. If these findings were indeed validated using another ILF2 perturbation condition, the data would suggest an impairment of the spare capacity of OXPHOS rather than a basal mitochondrial function.

The reviewer is totally correct. We have completely misinterpreted the initial reviewer's question. We apologize for this misunderstanding. Indeed, the results show an impairment of the spare capacity of OXPHOS rather than a basal mitochondrial function (**Figure 2**). We changed the text accordingly (page 10). Cells resistant to ILF2 depletion have an increased spare respiration

capacity, which suggests that these cells' have a higher adaptation capability to overcome stress conditions.

REVIEWERS' COMMENTS

Reviewer #1 (Remarks to the Author):

The authors have addressed all my concerns.

Reviewer #4 (Remarks to the Author):

In response to my previous comments on the Seahorse and OCR data (Fig. 4A), the authors provided additional seahorse experiments (rebuttal Fig 2A) using shRNAs. However, I remain unconvinced that the assay would suggest a robust change in OXPHOS. For instance, an increased basal OCR in NT ASOs + NSC compared to NT ASOs + Veh (Fig 4A), while a dramatic decrease basal OCR in NS sh + NSC compared to NS sh + Veh (rebuttal Fig 2A); and basal OCR is unchanged in ILF2 ASO (Fig 4A) while increased significantly in ILF2 NS (rebuttal Fig. 2A).

At this point, given the concerns on the Seahorse assay, perhaps the authors could reduce the emphasis in the text on the impact on mitochondrial bioenergetics and OXPHOS activation, but only focus on sensitizing the cells for ROS production, which was shown by MitoSOX and scRNAseq.

RESPONSE TO REVIEWERS' COMMENTS

Reviewer #4 (Remarks to the Author):

In response to my previous comments on the Seahorse and OCR data (Fig. 4A), the authors provided additional seahorse experiments (rebuttal Fig 2A) using shRNAs. However, I remain unconvinced that the assay would suggest a robust change in OXPHOS. For instance, an increased basal OCR in NT ASOs + NSC compared to NT ASOs + Veh (Fig 4A), while a dramatic decrease basal OCR in NS sh + NSC compared to NS sh + Veh (rebuttal Fig 2A); and basal OCR is unchanged in ILF2 ASO (Fig 4A) while increased significantly in ILF2 NS (rebuttal Fig. 2A).

At this point, given the concerns on the Seahorse assay, perhaps the authors could reduce the emphasis in the text on the impact on mitochondrial bioenergetics and OXPHOS activation, but only focus on sensitizing the cells for ROS production, which was shown by MitoSOX and scRNAseq.

We have followed the reviewer's suggestion and we have changed the title relative to Figure 4.